

# Influence of plant ecophysiology on ozone dry deposition: Comparing between multiplicative and photosynthesis-based dry deposition schemes and their responses to rising $CO_2$ level

Shihan Sun[1], Amos P. K. Tai[1,2], David H. Y. Yung[1], Anthony Y. H. Wong[1,3], Jason A. Ducker[4], and
Christopher D. Holmes[4]

[1]Earth System Science Programme and Graduate Division of Earth and Atmospheric Sciences, Faculty
of Science, The Chinese University of Hong Kong, Sha Tin, Hong Kong
[2]State Key Laboratory of Agrobiotechnology, and Institute of Environment, Energy and Sustainability,
The Chinese University of Hong Kong, Sha Tin, Hong Kong
[3]Department of Earth and Environmental, Boston University, Boston, USA
[4]Department of Earth, Ocean, and Atmospheric Science, Florida State University, Tallahassee, Florida,
USA

*Correspondence to*: Amos P. K. Tai (amostai@cuhk.edu.hk)

**Abstract.** Dry deposition is a key process for surface ozone ($O_3$) removal. Stomatal resistance is a
major component of $O_3$ dry deposition, which is parameterized differently in current land surface
models and chemical transport models. We developed and used a standalone terrestrial biosphere
model, driven by a unified set of prescribed meteorology, to evaluate two widely used dry deposition
modeling frameworks, Wesely (1989) and Zhang et al. (2003), with different configurations of stomatal
resistance: 1) the default multiplicative method in each deposition scheme; 2) the traditional
photosynthesis-based Farquhar-Ball-Berry (FBB) stomatal algorithm; 3) the Medlyn stomatal algorithm
based on an optimization theory. We found that using the FBB stomatal approach that captures
ecophysiological responses to environmental factors, especially to water stress, can generally improve
the simulated dry deposition velocities compared with multiplicative schemes. The Medlyn stomatal
approach produces higher stomatal conductance (reverse of stomatal resistance) than FBB and is likely
to overestimate dry deposition velocities for major vegetation types, but its performance is greatly
improved when spatially varying slope parameters based on annual mean precipitation are used. Large
discrepancies were also found in simulated stomatal responses to rising $CO_2$ levels, and that
multiplicative stomatal method with an empirical $CO_2$ response function produces reduction (–35%) in
global stomatal conductance, which is much larger than that with photosynthesis-based stomatal method
(–14–19%) when atmospheric $CO_2$ level increases from 390 ppm to 550 ppm. Our results show the
potential biases in $O_3$ sink caused by errors in model structure especially in the Wesely dry deposition
scheme, and the importance of using photosynthesis-based representation of stomatal resistance in dry
deposition schemes under a changing climate and rising $CO_2$ concentration.





# 1 Introduction

Tropospheric ozone ($O_3$) is a gaseous secondary air pollutant that is detrimental to human and vegetation health (Ainsworth et al., 2012; Fowler et al., 2009; Karnosky et al., 2007). Surface $O_3$ trends vary regionally over the recent decades, with reductions in Europe and North America and increases in many regions in Asia due to changes in anthropogenic emissions from industrial and agricultural processes (Cooper et al., 2014; Tarasick et al., 2019; Vingarzan, 2004). One of the major removal

pathways of tropospheric $O_3$ is dry deposition onto the land surface, accounting for ~25% of total tropospheric $O_3$ removal (Wild, 2007). Dry deposition of $O_3$ can be mainly divided into stomatal and non-stomatal deposition. In vegetated regions, stomatal $O_3$ uptake contributes ~45% of total $O_3$ dry deposition, which can cause potential injury to plant tissues and reduce plant productivity due to the oxidative nature of $O_3$ (Clifton et al., 2020a). Accurate representation of stomatal $O_3$ uptake is crucial

for near-surface $O_3$ modeling and $O_3$-induced damage assessment due to lack of correlation between stomatal $O_3$ flux and concentration (Ronan et al., 2020). Parameterization of dry deposition and its stomatal component remains to be one of the most unconstrained parts in tropospheric $O_3$ modeling, and models are still struggling to capture the observed spatiotemporal variations of $O_3$ dry deposition due to the complexity of dry deposition processes (Clifton et al., 2020a; Hardacre et al., 2015;

Stevenson et al., 2006; Young et al., 2018).

      Global chemical transport models (CTMs) typically employ the resistance-in-series model to compute dry deposition velocities of trace gases (e.g., Bey et al., 2001; Ching and Byun, 1999; Grell et al., 2005). Stomatal resistance is one of the major components of the resistance-in-series dry deposition schemes (Wesely and Hicks, 2000). The calculation of stomatal conductance (the reciprocal of

resistance) is also pivotal in the land surface component of Earth system models (ESMs) to quantify the partitioning of energy, water and carbon exchange between the land and atmosphere (Bonan, 2019; Sellers et al., 1996). Photosynthesis-based stomatal conductance has been implemented in various terrestrial biosphere or land surface models (LSMs) that are standalone or embedded within ESMs, but has rarely been used in CTMs to compute dry deposition rates; only few coupled climate-chemistry

models aiming to simulate climate-chemistry interactions have attempted to fully link dry deposition in the chemistry modules with photosynthesis in the land surface modules (e.g., Lei et al., 2020; Val Martin et al., 2014). Current CTMs typically use so-called "Jarvis-type", multiplicative stomatal resistance algorithms developed from Jarvis (1976), which apply semiempirical functions accounting for variations in environmental conditions to calculate dry deposition velocities (Emberson et al., 2000a;

Hicks et al., 1987; Meyers et al., 1998). Recent terrestrial biosphere models generally prefer photosynthesis-based approaches that link plant stomatal conductance directly to photosynthetic processes (Bonan, 2019). It has been suggested in recent studies that photosynthesis-based stomatal schemes that consider more sophisticated ecophysiological responses to environmental stimuli can improve the performance of CTMs in simulating dry deposition velocities (Lei et al., 2020; Otu-Larbi,

2021; Wu et al., 2011; Wong et al., 2019).

      Modeled $O_3$ dry deposition velocities and their dependence on stomatal behaviors have been evaluated in several recent studies (e.g., Lei et al., 2020; Lin et al., 2019; Szinyei et al., 2018; Wong et al., 2019). Regional and global CTMs can capture the seasonal variations and magnitudes of dry deposition fluxes within a factor of two (Hardacre et al., 2015; Silva and Heald, 2018). Uncertainties in



dry deposition modeling lie in various aspects such as incomplete knowledge of deposition processes, lack of long-term measurements, and insufficient accuracy in land use and vegetation characteristics (Clifton et al., 2020a). The traditional Wesely dry deposition scheme in CTMs usually applies a multiplicative stomatal resistance with a series of functions accounting for solar radiation, air temperature, seasonality, and biome type, without considering the effects of water stress on stomatal

uptake of $O_3$ or mechanistic representation of the plant ecophysiological responses to changing hydrometeorology and soil conditions (Wesely, 1989). Photosynthesis-based and some Jarvis-type multiplicative stomatal schemes are able to address these shortcomings with consideration of water stress, either explicitly via representation of water stress to plants, or via calibrated empirical water stress functions. Photosynthesis-based schemes have certain advantages over Jarvis-types schemes as

they parameterize the responses of plant stomata to environmental changes in a more mechanistic manner that explicitly accounts for the competing resource needs of plants with fewer empirical parameters (Franks et al., 2018; Medlyn et al., 2011). The Deposition of $O_3$ for Stomatal Exchange (DO$_3$SE) model uses a Jarvis-type stomatal algorithm with species-specific parameters to calculate stomatal $O_3$ deposition and predict $O_3$ damage for concerned tree and crop species in Europe (Büker et

al., 2015; Emberson et al., 2000a, 2000b, 2001, 2007). However, DO$_3$SE model was developed for species located in the boreal and temperate parts of Europe, and may not be easily generalizable to other species or plant types (Büker et al., 2015) or perform satisfactorily against site-specific data (e.g., Elvira et al., 2004). Jarvis-type and photosynthesis-based stomatal algorithms have been compared and evaluated in a few studies; photosynthesis-based schemes outperform multiplicative schemes in some

studies (e.g., Misson et al., 2004; Niyogi et al., 2009), but not in others (e.g., Büker et al., 2007; Uddling et al., 2005). Few studies have yet to compare or evaluate different stomatal approaches against global measurements under a fully consistent methodological framework with consistent model inputs. It is important to evaluate different types of stomatal algorithms thoroughly, not only to unify the representation of stomatal behaviors within ESMs for interactive land-atmosphere coupling, but also

to better represent plant-mediated processes that are relevant for atmospheric chemistry.

          Another motivation for better representation of plant-mediated processes in atmospheric chemistry modeling is to examine the potential influence of rising $CO_2$ levels under climate change, which can affect tropospheric $O_3$ concentrations through multiple ecological effects that modify the sources and sinks of tropospheric $O_3$, including $CO_2$ fertilization (Zhu et al., 2016), inhibition of

isoprene emission (Tai et al., 2013), and stomatal closure (Field et al., 1995). Changes in tropospheric $O_3$ concentrations can also be attributed to meteorological factors (e.g., sunlight, temperature, humidity, boundary layer stability, etc.) associated with $O_3$ chemistry and deposition processes (Camalier et al., 2007; Fowler et al., 2009; Kavassalis and Murphy, 2017). To explore climate change impacts on $O_3$ air quality, global CTMs concentrate on simulating the long-term effects of biogenic and anthropogenic

emission scenarios as well as atmospheric dynamics. Very few studies have addressed the ecological feedbacks due to lack of representations of biosphere-atmosphere interactions in CTMs (Centoni, 2017; Lei et al., 2020; Sanderson et al., 2007). Both stomatal and nonstomatal processes within plants play a role in observed $O_3$ dry deposition variations with changes in meteorology and atmospheric chemistry as suggested in recent studies (Clifton et al., 2020b; Knauer et al., 2020). However, the extent to which

plant stomata respond to rising $CO_2$ remains uncertain (Franks et al., 2013), impeding more accurate simulations under future climate scenarios. It is thus important to quantify how $O_3$ dry deposition may





respond to elevated $CO_2$ through stomatal regulation in order to better predict air quality and potential $O_3$ damage.

In this study, we examined whether or not $O_3$ dry deposition modeling in current CTMs can benefit from photosynthesis-based representation of stomatal resistance that has already been commonly used in terrestrial biosphere models. We first compared different dry deposition models that are commonly used at present. Modeled dry deposition velocity values were evaluated against globally distributed observations in different timescales for major land type categories. Multiplicative stomatal algorithms were compared with two photosynthesis-based stomatal conductance algorithms that have

been broadly implemented in terrestrial biosphere models, LSMs or coupled land-atmosphere models. The performance of different stomatal algorithms was also evaluated against ecosystem-level flux measurements on a global scale. We further discussed the importance of stomatal algorithm in dry deposition parameterization under elevated ambient $CO_2$ levels in atmospheric chemistry or air quality models.

**2 Data and methods**

**2.1 Model description**

For the numerical modeling framework, we made use of the Terrestrial Ecosystem Model in R (TEMIR), an offline ecosystem model driven by prescribed meteorology for investigating ecophysiological responses of the biosphere to atmospheric and environmental changes

(https://github.com/amospktai/TEMIR). This biosphere model has also been used in previous studies to evaluate global dry deposition fluxes (Wong et al., 2019) and the damage of ozone on global crop production (Tai et al., 2021). In this study, we implemented in TEMIR various representations of dry deposition velocity and stomatal resistance in particular. The dry deposition parameterization schemes are all based on the big-leaf representation of the terrestrial biosphere. We examined two major dry

deposition modeling frameworks: (1) the Wesely framework (referred to as W89), which has been widely used in global atmospheric chemistry models (Hardacre et al., 2015; Morgenstern et al., 2017), and in this study we used the W89 framework as currently implemented in the GEOS-Chem chemical transport model (Wang et al., 1998); (2) the Zhang et al. (2003) dry deposition framework (referred to as Z03) used in several regional air quality models (Nopmongcol et al., 2012; Schwede et al., 2011;

Zhang et al., 2009). Under both the W89 and Z03 frameworks, dry deposition velocity ($v_d$) is calculated as the inverted sum of aerodynamic resistance ($R_a$), quasi-laminar sublayer resistance ($R_b$), and bulk surface resistance ($R_c$) following

$$v_d = \frac{1}{R_a + R_b + R_c} , \qquad\qquad (1)$$

$R_a$ is controlled by micrometeorological conditions and the surface roughness, and is calculated based

on the Monin-Obukhov similarity theory (Monin and Obukhov, 1954) with the stability function from Foken (2006). $R_b$ is a function of friction velocity ($u_*$) and molecular diffusivity (Wesely and Hicks, 1977). $R_a$ and $R_b$ in different models are generally computed with similar methods, while the calculation of $R_c$ differs the most. Here we used the same parameterization of $R_a$ and $R_b$ for both Z03 and W89 to





focus on model discrepancies that could arise from $R_c$. The term $R_c$ is generally calculated as a series of
parallel resistances including stomatal resistance ($R_s$), cuticular resistance ($R_{cut}$), and ground resistance
($R_g$). Details of each term are presented in Table 1. Here we mainly focused on the influence of different
stomatal resistance representations on dry deposition velocity of $O_3$ ($v_d$), and compared the differences
among them; we did so by implementing not only the default multiplicative $R_s$ schemes in W89 and
Z03, but also photosynthesis-based $R_s$ schemes, as described below.


**Table 1. Description of dry deposition configurations used in this study.**

| | W89 | W89FBB | W89MED | Z03 | Z03FBB | Z03MED |
|---|---|---|---|---|---|---|
| $R_a$ | Stable conditions: $R_a = \frac{1}{\kappa u_*}(\log(\frac{z}{z_0}) + 5\frac{z-z_0}{L})$ <br> Unstable conditions: $R_a = \frac{1}{\kappa u_*}(\log(\left\|\frac{\sqrt{1-15z/L}-1}{\sqrt{1-15z/L}+1}\right\|) - \log(\left\|\frac{\sqrt{1-15z_0/L}-1}{\sqrt{1-15z_0/L}+1}\right\|))$ | | | | | |
| $R_b$ | $R_b = \frac{2}{\kappa u_*}(S_c/P_r)^{0.667}$ | | | | | |
| $R_s$ | Eq. (2) | Eq. (4) | Eq. (5) | Eq. (3) | Eq. (4) | Eq. (5) |
| $R_c$ | $\frac{1}{R_c} = \frac{1}{R_s} + \frac{1}{R_{cut}} + \frac{1}{R_{adc} + R_{clx}} + \frac{1}{R_g + R_{ag}}$ | | | $\frac{1}{R_c} = \frac{1 - W_{st}}{R_s} + \frac{1}{R_{cut}} + \frac{1}{R_{ac} + R_g}$ | | |
| $R_{cut}$ | $R_{cut} = \begin{cases} \frac{R_{cut0}}{LAI} + 1000e^{-T-4}, & T \geq -1 \\ R_{cut0} \times \min(2, e^{0.2(-1-T)}), & T < -1 \end{cases}$ | | | For dry canopies: $R_{cutd} = \frac{R_{cutd0}}{e^{0.03RH}LAI^{1/4}u_*}$ <br> For wet canopies: $R_{cutw} = \frac{R_{cutw0}}{LAI^{0.5}u_*}$ | | |
| $R_{adc}$ and $R_{ac}$ | $R_{adc} = 100(1 + \frac{1000}{SRAD + 10})$ | | | $R_{ac} = \frac{R_{ac0}LAI^{1/4}}{u_*^2}$ <br> $R_{ac0}(t) = R_{ac0}(min) + \frac{LAI(t) - LAI(min)}{LAI(max) - LAI(min)}$ | | |
| $R_g$ and $R_{ag}$ | Prescribed | | | Prescribed for wet and dry surfaces | | |

\* $\kappa$ = von Kármán constant; $u_*$ = friction velocity; $z_0$ = roughness height; $z$ = reference height; $L$ =
Obukhov length; $S_r$ = the Schmidt number; $P_r$ = the Prandtl number for air; LAI = leaf area index; $T$ =
surface temperature (C°); SRAD: incoming shortwave solar radiation; $R_c$ = canopy resistance; $R_{cut}$ =
cuticular resistance; $R_{adc}$ = lower canopy aerodynamic resistance; $R_{clx}$ = lower canopy resistance; $R_g$ =
ground resistance; $R_{ag}$ = ground aerodynamic resistance; $R_{ac}$ = in-canopy aerodynamic resistance; $W_{st}$ =
stomatal blocking factor; RH = relative humidity; $R_{cutd0}$ and $R_{cutw0}$ = reference cuticular resistance for
dry and wet conditions.

In this study, we focus on comparing the different representations of $R_s$ in dry deposition
schemes. Different from the Jarvis-type algorithms commonly used in calculating dry deposition
velocities, terrestrial biosphere models generally prefer the photosynthesis-based parameterization in
order to calculate transpiration rate and carbon uptake. The Ball-Woodrow-Berry (BWB) model (Ball et
al., 1987), which describes an empirical relationship between stomatal conductance, photosynthesis
rate, RH and leaf-surface $CO_2$ concentration, was integrated with the Farquhar et al. (1980)
photosynthesis model (collectively referred to as the Farquhar-Ball-Berry model, or FBB, hereafter) and
introduced to terrestrial biosphere models in order to quantify ecosystem fluxes to and from the
atmosphere starting from the mid-90s (Sellers et al., 1996). Medlyn et al. (2011) proposed a stomatal
conductance model (referred to as MED hereafter) based on the theory whereby plants optimize its



stomatal behavior so as to maximize photosynthesis for given water availability (Cowan and Farquhar, 1977). MED has been parameterized with a global leaf-level gas exchange database (Lin et al., 2015), and recently implemented to replace FBB in some global land surface models (Haverd et al., 2018; Lawrence et al., 2019). The potential of implementing the optimal theory in stomatal conductance models has also been emphasized in many recent studies as they can provide a more theoretical explanation to model parameters and thus a higher predictive power under changing environments (Bai et al., 2019; Buckley et al., 2017; Katul et al., 2010; Lu et al., 2016; Sperry et al., 2017).

We examined and compared four representative stomatal schemes, including the default parameterizations in W89 and Z03, as well as two photosynthesis-based stomatal conductance ($g_s$) modules FBB and MED. The default stomatal resistance scheme in W89 is as follows:

$$R_s = 1/[G_s(\text{LAI}, \text{PAR})f(T)D_i/D_v], \tag{2}$$

where $G_s(\text{LAI}, \text{PAR})$ represents dependence of canopy stomatal conductance on LAI and on direct and diffuse photosynthetically active radiation (PAR) within canopy as described in Wang et al. (1998). $f(T)$ represents the temperature effects on stomatal resistance. $D_i$ and $D_v$ are molecular diffusivities for water and the pollutant gas respectively. Details of Eq. (2) is described in Supplementary Text S2. The default Z03 stomatal resistance scheme follows a two-big-leaf canopy resistance model developed by Hicks et al. (1987):

$$R_s = 1/[G_s(\text{LAI}, \text{PAR})f(T)f(\text{VPD})f(\psi)D_i/D_v], \tag{3}$$

where $G_s(\text{LAI}, \text{PAR})$ represents unstressed total canopy stomatal conductance calculated by summing the contribution from sunlit and shaded leaves. $f(T)$, $f(\text{VPD})$ and $f(\psi)$ are dimensionless stress functions for temperature ($T$), vapor pressure deficit (VPD), and water stress ($\psi$) respectively, as described in Brook et al. (1999). These stress functions take different forms, and their details are described in the Supplementary Text S2.

Both FBB and MED employ the Ball-Berry approach that links leaf photosynthesis with stomatal conductance. The FBB stomatal conductance scheme computes leaf stomatal resistance as follows:

$$g_s = \frac{1}{r_s} = g_{1B}\frac{A_n h_s}{C_s} + g_0, \tag{4}$$

where $A_n$ is leaf net photosynthesis (μmol $CO_2$ m$^{-2}$ s$^{-1}$), $h_s$ is leaf surface relative humidity, $C_s$ is $CO_2$ concentration at the leaf surface (μmol mol$^{-1}$), and $g_{1B}$ is the fitted slope parameter. $g_0$ is PFT-dependent minimum stomatal conductance (μmol $CO_2$ m$^{-2}$ s$^{-1}$). The MED stomatal scheme is implemented as described in Medlyn et al. (2011):

$$g_s = \frac{1}{r_s} = 1.6(1 + \frac{g_{1M}}{\sqrt{\text{VPD}}})\frac{A_n}{C_s} + g_0, \tag{5}$$

where $g_{1M}$ is similar to $g_{1B}$ as above. The prescribed parameters $g_0$ and $g_{1M}$ are from Lin et al. (2015). For MED and FBB, leaf stomatal conductance is coupled to photosynthetic rate, calculated for sunlit and shaded leaves respectively, and then scaled up to the canopy level. Canopy stomatal conductance ($G_s$) is calculated as:





$$G_{\mathrm{s}} = \frac{1}{R_{\mathrm{s}}} = \left(\frac{1}{r_{\mathrm{b}}+r_{\mathrm{s}}^{\mathrm{sun}}} L^{\mathrm{sun}} + \frac{1}{r_{\mathrm{b}}+r_{\mathrm{s}}^{\mathrm{sha}}} L^{\mathrm{sha}}\right) D_{\mathrm{i}}/D_{\mathrm{v}}, \tag{6}$$

where $r_{\mathrm{s}}^{\mathrm{sun}}$ and $r_{\mathrm{s}}^{\mathrm{sha}}$ are sunlit and shaded stomatal resistance respectively, $L^{\mathrm{sun}}$ and $L^{\mathrm{sha}}$ are sunlit and shaded LAI respectively, $r_{\mathrm{b}}$ is leaf boundary resistance. Details of the photosynthesis-stomatal conductance module in TEMIR is also described in the Supplementary Text S2.

220  Differences between the W89 and Z03 frameworks lie in not only stomatal parameterization, but also non-stomatal deposition structures and algorithms. The W89 here represents the default in GEOS-Chem that is extensively used (e.g., Hardacre et al., 2015; Porter et al., 2019; Silva and Heald, 2018). To evaluate the two dry deposition frameworks and to compare the multiplicative and photosynthesis-based stomatal schemes, we replaced the default stomatal parameterization in W89 and Z03 dry
225 deposition frameworks with FBB and MED, and in total six dry deposition configurations were tested as described in Table 1.

  Simulations using each dry deposition configuration were conducted in the single-site mode for the observational sites listed in Supplementary Table S1. For most of the simulations, we used reanalyzed meteorological data from the Modern-Era Respective analysis for Research and Applications
230 version 2 (MERRA-2) (Gelaro et al., 2017), which provides all the required meteorological input data for simulations. We also directly used the standard meteorological data from FLUXNET2015 dataset (Pastorello et al., 2020) to replace the default MERRA-2 data for FLUXNET observational sites. Cloud fraction and soil moisture data were provided by MERRA-2 for all sites. Observed site-specific LAI values were obtained from the references listed in Table S1. We applied regridded Moderate Resolution
235 Imaging Spectroradiometer (MODIS) LAI for sites where site-specific LAI data were not available. For most of the soil and plant parameters required for TEMIR simulations, we used the Community Land Model version 4.5 (CLM4.5) land surface dataset (Lawrence and Chase, 2007) that provides parameters specific for different plant functional types (PFTs). CLM4.5 land types were mapped with W89 and Z03 land types as described in Table S3. For global simulations, the model was run at a spatial resolution of
240 2°×2.5° driven by MERRA-2 meteorology for each dry deposition configuration.

## 2.2 Field measurements

  We compared our model results to the aggregated observations from 41 datasets of direct measurements of $O_3$ flux and $v_{\mathrm{d}}$ (Hardacre et al., 2015; Silva et al., 2018; Lin et al., 2019). All datasets used here were obtained with the eddy covariance (EC) method (Baldocchi et al., 1988). The
245 observational sites we used covered five major vegetation types: deciduous broadleaf forest (DBF), evergreen needleleaf forest (ENF), crop (CRO), grass (GRA), and tropical rainforest (TRF), and the majority of sites were concentrated in the US and Europe from short-term projects. Modeled seasonal mean $v_{\mathrm{d}}$ values are evaluated against this compilation of observational datasets in the following section. A more detailed description of observational datasets and the corresponding references are also listed in
250 the Supplementary Table S1.

  To further evaluate model capability in capturing diurnal $v_{\mathrm{d}}$ and $G_{\mathrm{s}}$, we investigated four long-term observational sites listed in Table 2: Harvard Forest, Hyytiälä Forest, Borden Forest and Blodgett Forest. These four sites provided continuous EC measurements for momentum, sensible heat, latent heat





and $O_3$ fluxes on an hourly basis for more than five years. Details of each long-term site and their data
filtering methods are described in Supplementary Text S1. Canopy stomatal conductance values at the
long-term measurement sites were estimated based on the inverted Penman-Monteith method (referred
to as P-M hereafter) using site-level FLUXNET meteorological measurements (Gerosa et al., 2007).
Stomatal conductance of $O_3$ was then calculated using molecular diffusion coefficient ratio between $O_3$
and water vapor.

**Table 2. Description of long-term $O_3$ flux measurements.**

|  | latitude | longitude | Data period | Vegetation | Reference |
|---|---|---|---|---|---|
| **Harvard Forest** | 42.7ºN | 72.2ºW | 1991~2009 | Red oak, red maple | Munger et al. (1996) |
| **Hyytiälä Forest** | 61.85ºN | 24.28ºE | 2005~2016 | Scots pine | Keronen et al. (2003) |
| **Blodgett Forest** | 38.9ºN | 120.6ºW | 2001~2007 | Ponderosa pine | Fares et al. (2010) |
| **Borden Forest** | 44.3 ºN | 79.9 ºW | 2008~2013 | Red maple, white pine, large-tooth aspen | Wu et al. (2018) |

To evaluate simulated $G_s$ with different stomatal conductance algorithms on a larger
spatiotemporal scale, we utilized the recent dataset of SynFlux
(https://doi.org/10.5281/zenodo.1402054) that provides monthly daytime $G_s$ calculated with the P-M
method using standard micrometeorological flux measurements at 103 FLUXNET sites concentrated in
the US and Europe where $O_3$ monitoring networks are available (Ducker et al., 2018). We applied
FLUXNET meteorology and MODIS LAI for simulations at FLUXNET sites. Simulated average
monthly daytime $G_s$ during the measurement periods were compared with SynFlux $G_s$ for each
FLUXNET site. The uncertainties in $G_s$ due to the fraction of soil evaporation in evapotranspiration
measurements were restricted with filtered data as described in Ducker et al. (2018). The definition of
daytime follows that in Ducker et al. (2018) (i.e., solar elevation angle above 4º) for comparison with
SynFlux $G_s$.

## 3 Comparison and evaluation with observations

### 3.1 Evaluation of simulated seasonal average $v_d$

The simulated seasonal average daytime $v_d$ using different dry deposition schemes in Table 1
were evaluated with observations for major PFTs. We used two unbiased symmetric metrics: the
normalized mean bias factor (NMBF) and normalized mean absolute error factor (NMAEF), to evaluate
different dry deposition schemes (Yu et al., 2006). Positive NMBF values are interpreted as
overestimation by a factor of 1 + NMBF, while negative NMBF means underestimation by a factor of 1
– NMBF. Smaller absolute values of NMBF and NMAEF indicate better agreement with observations.
Seasonal daytime mean observed and simulated $v_d$ and NMBF values are summarized for five major
PFTs (DBF: Deciduous Broadleaf Forest, ENF: Evergreen Needleleaf Forest, CRO: Crop, TRF:
Tropical Rainforest; GRA: Grass) in Table 3.





**Table 3. Seasonal mean and standard deviation of the observed and simulated ozone dry deposition velocity (cm/s) (daytime: LT 6:00-18:00). DBF: Deciduous Broadleaf Forest; ENF: Evergreen Needleleaf Forest; CRO: Crop; TRF: Tropical Rainforest; GRA: Grass.**

| PFT | Season | Observation | W89 | | | W89FBB | | | W89MED | | | Z03 | | | Z03FBB | | | Z03MED | | |
|---|---|---|---|---|---|---|---|---|---|---|---|---|---|---|---|---|---|---|---|---|
| | | mean±sd | mean±sd | NMBF | NMAEF | mean±sd | NMBF | NMAEF | mean±sd | NMBF | NMAEF | mean±sd | NMBF | NMAEF | mean±sd | NMBF | NMAEF | mean±sd | NMBF | NMAEF |
| DBF | JJA | 0.69±0.10 | 0.90±0.17 | 0.32 | 0.32 | 0.59±0.10 | -0.16 | 0.26 | 0.81±0.24 | 0.18 | 0.41 | 0.55±0.09 | -0.25 | 0.30 | 0.58±0.11 | -0.18 | 0.26 | 0.78±0.25 | 0.14 | 0.41 |
| | MAM | 0.33±0.02 | 0.42±0.13 | 0.27 | 0.43 | 0.28±0.08 | -0.21 | 0.23 | 0.35±0.10 | 0.05 | 0.26 | 0.29±0.08 | -0.13 | 0.27 | 0.31±0.05 | -0.09 | 0.18 | 0.37±0.08 | 0.10 | 0.21 |
| | SON | 0.52±0.20 | 0.49±0.12 | -0.05 | 0.18 | 0.29±0.07 | -0.78 | 0.78 | 0.39±0.13 | -0.34 | 0.34 | 0.41±0.06 | -0.26 | 0.26 | 0.37±0.05 | -0.39 | 0.39 | 0.46±0.11 | -0.11 | 0.13 |
| | DJF | 0.25±0.08 | 0.14±0.05 | -0.86 | 0.97 | 0.14±0.05 | -0.86 | 0.86 | 0.15±0.06 | -0.72 | 0.87 | 0.24±0.04 | -0.04 | 0.21 | 0.26±0.03 | 0.02 | 0.23 | 0.26±0.04 | 0.05 | 0.27 |
| ENF | JJA | 0.58±0.23 | 0.46±0.12 | -0.29 | 0.35 | 0.46±0.11 | -0.30 | 0.42 | 0.47±0.10 | -0.27 | 0.40 | 0.42±0.14 | -0.39 | 0.68 | 0.52±0.14 | -0.14 | 0.44 | 0.53±0.13 | -0.12 | 0.42 |
| | MAM | 0.46±0.15 | 0.35±0.11 | -0.31 | 0.43 | 0.34±0.10 | -0.34 | 0.40 | 0.37±0.12 | -0.24 | 0.37 | 0.42±0.06 | -0.10 | 0.31 | 0.43±0.09 | -0.07 | 0.26 | 0.46±0.11 | -0.01 | 0.26 |
| | SON | 0.47±0.22 | 0.35±0.12 | -0.35 | 0.43 | 0.28±0.07 | -0.64 | 0.68 | 0.26±0.04 | -0.83 | 0.85 | 0.39±0.13 | -0.21 | 0.46 | 0.41±0.15 | -0.13 | 0.37 | 0.40±0.12 | -0.18 | 0.43 |
| | DJF | 0.32±0.21 | 0.17±0.07 | -0.87 | 0.89 | 0.19±0.08 | -0.66 | 0.73 | 0.16±0.06 | -0.98 | 1.01 | 0.30±0.11 | -0.08 | 0.29 | 0.30±0.15 | -0.05 | 0.28 | 0.28±0.12 | -0.14 | 0.36 |
| CRO | / | 0.53±0.16 | 0.50±0.26 | -0.05 | 0.29 | 0.72±0.15 | 0.37 | 0.43 | 0.81±0.13 | 0.54 | 0.54 | 0.54±0.11 | 0.03 | 0.18 | 0.61±0.15 | 0.16 | 0.32 | 0.67±0.14 | 0.27 | 0.32 |
| TRF | / | 0.76±0.48 | 1.11±0.07 | 0.46 | 0.56 | 0.98±0.06 | 0.29 | 0.52 | 1.10±0.10 | 0.44 | 0.53 | 0.47±0.05 | -0.60 | 0.85 | 0.57±0.04 | -0.33 | 0.61 | 0.66±0.07 | -0.14 | 0.48 |
| GRA | JJA | 0.33±0.17 | 0.72±0.10 | 1.21 | 1.21 | 0.59±0.21 | 0.82 | 0.82 | 0.84±0.28 | 1.56 | 1.56 | 0.50±0.12 | 0.53 | 0.79 | 0.50±0.16 | 0.51 | 0.51 | 0.68±0.21 | 1.08 | 1.08 |
| | MAM | 0.39±0.13 | 0.58±0.13 | 0.48 | 0.48 | 0.43±0.00 | 0.08 | 0.28 | 0.62±0.16 | 0.57 | 0.74 | 0.42±0.11 | 0.06 | 0.48 | 0.46±0.03 | 0.17 | 0.36 | 0.62±0.15 | 0.56 | 0.72 |
| | SON | 0.30±0.06 | 0.59±0.21 | 1.00 | 1.20 | 0.46±0.22 | 0.55 | 0.78 | 0.55±0.26 | 0.88 | 1.03 | 0.42±0.29 | 0.43 | 0.76 | 0.46±0.20 | 0.54 | 0.80 | 0.54±0.22 | 0.82 | 0.95 |
| | DJF | 0.33±0.05 | 0.34±0.26 | 0.02 | 0.68 | 0.24±0.14 | -0.37 | 0.56 | 0.34±0.31 | 0.04 | 0.77 | 0.31±0.15 | -0.08 | 0.46 | 0.35±0.18 | 0.06 | 0.49 | 0.443±0.32 | 0.31 | 0.79 |


Figure 1 shows the comparison between simulated and observed daytime (6:00am~18:00pm) average $v_d$ for five major PFTs categorized by seasons. The six dry deposition schemes can generally capture the magnitude of seasonal daytime $v_d$ for major PFTs. For deciduous forests as shown in Fig. 1a, Z03 underestimates $v_d$ in general (NMBF = –0.20; NMAEF = 0.28), while W89 overestimates $v_d$

(NMBF = 0.19; NMAEF = 0.31), with positive biases especially during summer. For coniferous forests, W89 and Z03 underestimate observed $v_d$ as shown in Fig. 1b. Both W89FBB and W89MED produce higher positive biases in simulated daytime $v_d$ compared with W89 for deciduous and coniferous forests, while Z03FBB and Z03MED can reproduce observed $v_d$ with lower NMBF values than Z03. Negative biases simulated with Z03 can be caused by the prescribed maximum canopy stomatal conductance for

coniferous forest, which was set lower than deciduous forest. More recent studies have observed higher unstressed maximum stomatal conductance for coniferous forest than deciduous forest (Hoshika et al., 2017). Figure 1c shows that for grasses, all dry deposition schemes overestimate $v_d$, while Z03 and Z03FBB generally produce lower mean absolute biases (NMAEF < 0.4) than other deposition schemes. In previous works, models underestimated grassland $v_d$ mostly (Hardacre et al., 2015; Pio et al., 2000).

Modeled $v_d$ for grasses is largely determined by the prescribed minimum stomatal resistance ($r_{smin}$) and LAI. For example, Pio et al. (2000) used $r_{smin}$ (1200 s m$^{-1}$) that is higher than the value (200 s m$^{-1}$) we used in W89 and Z03 for grasses, resulting in lower simulated $v_d$ values than ours. In Hardacre et al. (2015), W89 underestimated $v_d$ at a long-term moorland site using $r_{smin}$ (200 s m$^{-1}$) and prescribed MODIS LAI, whereas in our study for the evaluation of this particular site, W89 overestimates daytime

$v_d$ with positive biases of about 0.3 cm s$^{-1}$ using observed LAI provided in Flechard and Fowler (1998). Observed LAI values for the observational grassland sites used in this study are higher than the grid-level MODIS LAI in the corresponding grid cell, leading to discrepancies in modeled $v_d$.





**Figure 1. Average daytime (LT 6:00~18:00) observed and simulated dry deposition velocities for five land types. Each data point refers to seasonal average daytime O₃ dry deposition velocity from one dataset listed in Table S1. Colors indicate dominant seasons during field measurements, except that for crops different colors indicate crop types (C3 and C4 crops).**

For crops as shown in Fig. 1d, Z03 better reproduces $v_d$ than the other deposition schemes with lower mean biases (NMBF = 0.01; NMAEF = 0.19). For rainforests shown in Fig. 1e, all dry deposition schemes simulate nearly constant daytime $v_d$ values for different sites, which is mainly due to the



relatively uniform LAI input and meteorological conditions for tropical regions during the measurement periods. The source of discrepancies between model and observations is not clear, which can arise from various aspects such as leaf age stage of tropical trees, as well as uncertainties in the flux measurement themselves. Canopy storage effects can mask observed diurnal $O_3$ deposition variations as previously
found (Rummel et al., 2007). Current $O_3$ flux measurements for rainforests are rather limited especially for the dry season, which also prohibits precise model parameterization (Fan et al., 1990; Sigler et al., 2002). Non-stomatal $O_3$ deposition includes chemical reactions from soil emissions of nitric oxide (NO) together with biogenic volatile organic compounds (BVOC) (Fares et al., 2012). Recent studies have also found that in tropical rainforests, strong sources of sesquiterpenes are emitted from soil and can
react rapidly with $O_3$, contributing to non-stomatal deposition that is previously unreported (Bourtsoukidis et al., 2018).

We also compared nighttime $v_d$ simulated with different deposition schemes as shown in Supplementary Fig. S1. Simulated nighttime $G_s$ is close to zero and thus modeled $O_3$ dry deposition velocity mainly consists of non-stomatal sink. Field measurements have shown that non-stomatal
deposition is not negligible throughout the day, and that non-stomatal deposition velocity can have diurnal cycles similar to that of $G_s$ with even higher deposition rates during the day (Hogg et al., 2007). Observed nighttime $G_s$ is generally minimal, lower than non-stomatal conductance over vegetated regions (Caird et al., 2007; Hogg et al., 2007). W89 underestimates nighttime $v_d$ with large negative biases (NMBF < –1.4) for both deciduous and coniferous forests primarily due to underestimated non-
stomatal deposition. This systematic negative bias in non-stomatal deposition can also induce misrepresentation of stomatal and non-stomatal partitioning during the day.

Overall, W89 and Z03 with multiplicative stomatal approaches produce similar biases, yet biases from Z03 is generally slightly smaller than W89 when evaluated with observations on a seasonal timescale. We found that Z03FBB generally produces lower biases, with Z03 non-stomatal
parameterization and photosynthesis-based FBB stomatal conductance. Replacing the default multiplicative stomatal approach in W89 and Z03 with photosynthesis-based MED stomatal parameterization can induce higher absolute biases in simulated daytime $v_d$.

### 3.2 Comparison of simulated diurnal $v_d$ at long-term measurement sites

We also evaluated simulated seasonal and diurnal $v_d$ variations using different dry deposition
schemes at four long-term measurement sites listed in Table 2. Meteorological variables of temperature, relative humidity (RH), vapor pressure deficit (VPD), and root-zone soil wetness (SW) at selected sites are summarized in Supplementary Table S2. Figure 2 shows observed and simulated monthly daytime (6:00am~18:00pm) $v_d$ at each long-term site. Highest $v_d$ values are typically observed in summer (JJA), during which large discrepancies are also found between modeled and observed $v_d$ values. Therefore,
we focused on summertime months when highest levels of $O_3$ concentrations and $v_d$ co-occur. W89, W89FBB, and W89MED overestimate monthly daytime $v_d$ with higher positive biases than Z03, Z03FBB and Z03MED at Harvard Forest and Borden Forest during growing seasons. At Hyytiälä Forest, no specific scheme can better capture $v_d$ than the others. At Blodgett forest, all dry deposition





schemes underestimate $v_d$ values during JJA. We further examined simulated diurnal cycles of $v_d$ and $G_s$
to analyze the performances of different dry deposition schemes in the following in this section.

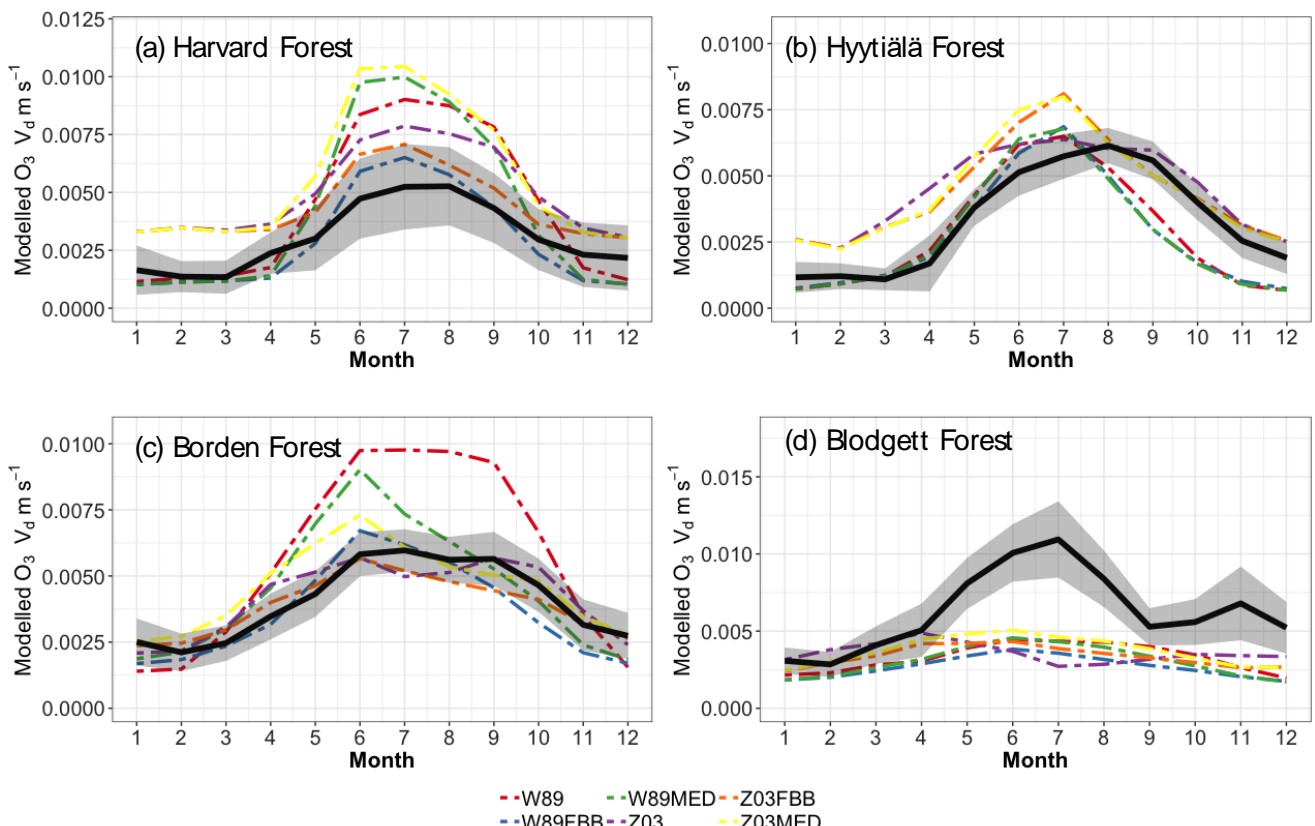

**Figure 2. Average monthly daytime (local time 06:00~18:00) ozone dry deposition velocity. Black solid lines indicate observed average monthly $v_d$. Shaded envelope shows standard deviation of observed summertime average monthly daytime $v_d$. Colored lines indicate simulated average monthly $v_d$ using different dry deposition schemes.**


Figure 3 shows modeled and observed JJA diurnal $v_d$ cycles. Overall, the diurnal cycle is characterized by a sharp early morning rise in $v_d$, followed by a gentle decline throughout the day (sometimes with a midday dip) and finally by a steeper decline toward early evening; such a typical shape strongly resembles the drawing of "a boa constrictor digesting an elephant" in the famous novella
*The Little Prince*. Most of the schemes can capture this typical shape, with the notable exception of W89, which simply reflects a symmetric function of solar zenith angle. At Harvard Forest shown in Fig. 3a, Z03, W89FBB and Z03FBB can well reproduce the average diurnal cycle of $v_d$, while W89MED and Z03MED overestimate $v_d$ with early morning peaks, and W89 overestimates it with a peak shifted later in the day. Figure 4 shows the modeled and observed diurnal $G_s$ cycles at the four sites calculated
with the P-M method. As shown in Fig. 4a, overestimated $v_d$ values by W89MED and Z03MED are primarily caused by the positive biases in simulated $G_s$ peaks during early morning and late afternoon.





W89 overestimates summertime $v_d$, which is mainly caused by overestimated afternoon $v_d$. Previous studies have also found that the Wesely scheme overestimated $v_d$ at Harvard Forest, and assumed that the positive biases were caused by overestimated LAI from satellite observations (Hardacre et al., 2015; Silva and Heald, 2018). However, the overestimation of $v_d$ mostly arises from model parameterization as we used observed site-level LAI values in this study. Stomatal deposition dominates over non-stomatal deposition at Harvard Forest in summer during the day (Fig. S3). Overestimated $v_d$ at Harvard Forest is mainly caused by the stomatal parameterization, which is also emphasized in the evaluation of $G_s$ and global simulations in the following sections of this study.

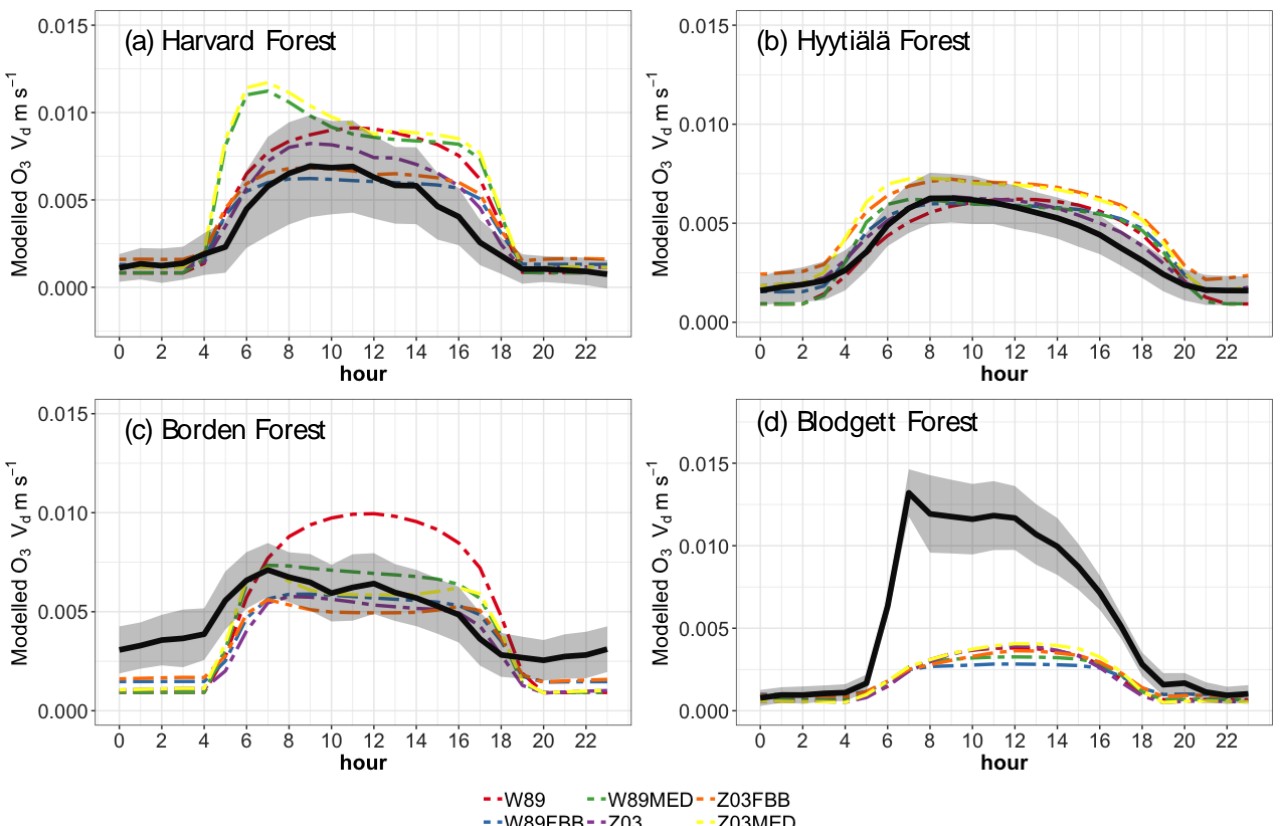

**Figure 3.** Average diurnal cycles of ozone dry deposition velocity at four long-term observational sites. Black solid lines indicate observed average diurnal cycles of $v_d$. Shaded envelope indicates standard deviation of summertime average hourly $v_d$. Dashed lines indicate simulated diurnal cycles of $v_d$ using different dry deposition schemes.



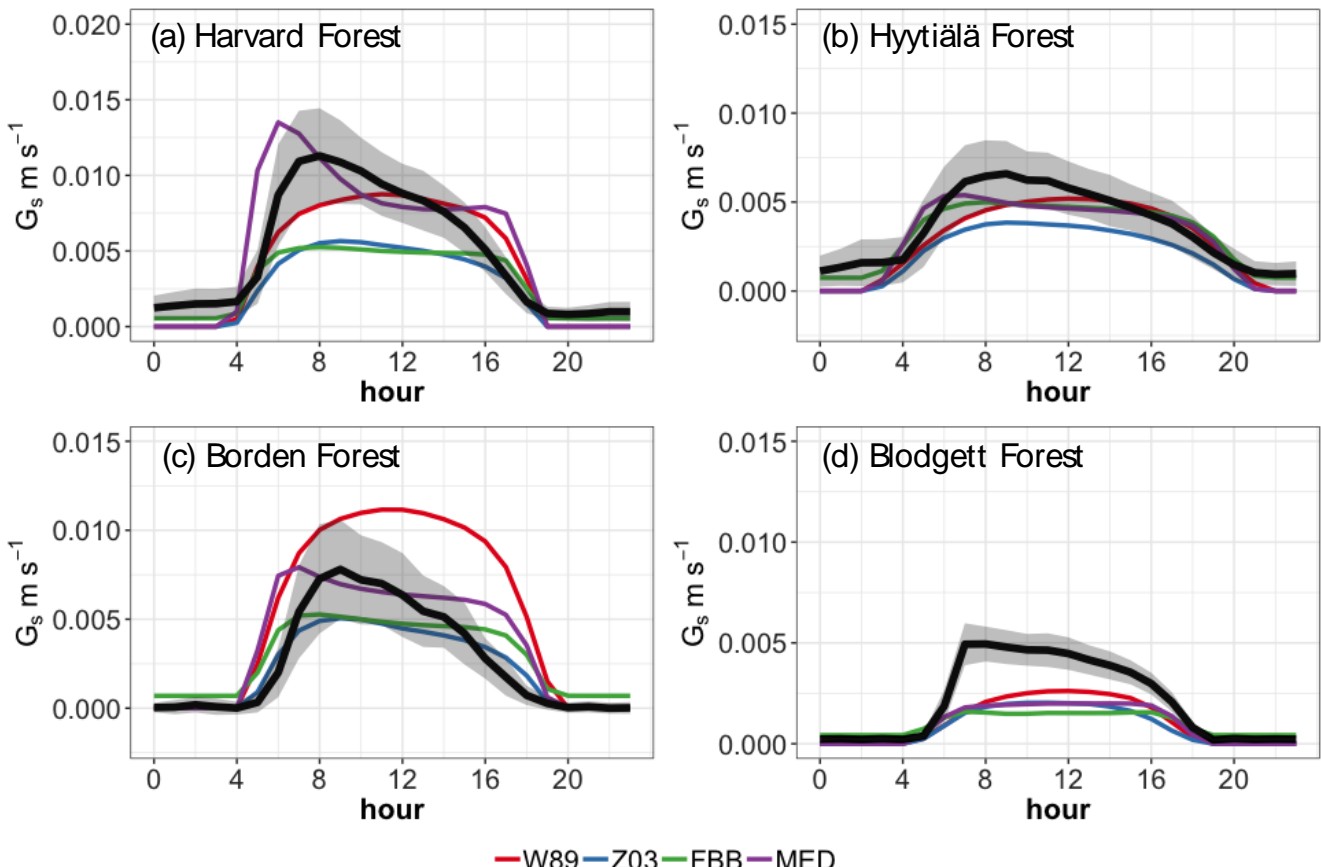

**Figure 4. Simulated and observed diurnal cycles of canopy stomatal conductance for ozone during summer at the four long-term measurement sites. Black lines indicate $G_s$ derived with P-M method. Shaded envelope shows standard deviation of summertime average hourly $G_s$. Colored solid lines indicate simulated stomatal conductance using multiplicative and photosynthesis-based stomatal approaches.**


The diurnal $v_d$ variations at Hyytiälä Forest can be well captured by different dry deposition schemes shown in Figure 3b. Again, W89 does not capture the typical "boa" shape. However, as shown in Figure 3d, different dry deposition schemes underestimate $v_d$ at Blodgett Forest with large negative

biases despite that Hyytiälä Forest and Blodgett Forest are both pine-dominated forests. The major $O_3$ removal process in ponderosa pine forest is non-stomatal $O_3$ sink through in-canopy chemical reactions between $O_3$ and BVOC (Fares et al., 2010; Kurpius and Goldstein, 2003). Rannik et al. (2012) analyzed the partitioning between stomatal and non-stomatal $O_3$ deposition at Hyytiälä Forest, finding that $O_3$ gas-phase chemistry is not the major contributor to $O_3$ removal during the day. Different meteorological

conditions at these two pine forest sites also result in discrepancies in simulated $v_d$. The Blodgett forest site is characterized by a Mediterranean climate with high surface temperature and VPD during the day for the simulation period (Table S2). Hyytiälä Forest is located in a boreal region lower surface temperature and VPD than Blodgett Forest during summer. Previous studies have also found an exponential dependence of non-stomatal $O_3$ deposition rates on temperature through gas-phase reactions





with biogenic hydrocarbons in ponderosa pine forests (Kurpius and Goldstein, 2003). Figure 4d shows
that different stomatal conductance schemes struggle to capture the magnitude of daytime stomatal $O_3$
sink at Blodgett Forest where water supply is limited. Besides misrepresentation of non-stomatal
deposition as discussed above, underestimation of total $O_3$ dry deposition can also be caused by not
accounting for BVOC ozonolysis and non-transpiring surface deposition in dry deposition schemes.

415            For Borden Forest as shown in Figure 3c, models can well capture observed $v_d$, except that W89
overestimates $v_d$ and does not capture the typical "boa" shape. Positive biases in W89-simulated $v_d$ is
mainly caused by overestimated afternoon $G_s$ (Fig. 4c), considering that stomatal sink dominates total
$O_3$ dry deposition at Borden Forest (Fig. S3). However, underestimation of JJA $v_d$ at Borden Forest has
been found in WRF-Chem simulations, which also applied the Wesely scheme (Wu et al., 2018). In our
study, the W89 scheme with modification by Wang et al. (1998) applies a function for light adjustment
on $R_s$ using solar radiation and LAI, while in the Wesely scheme within WRF-Chem, LAI is not
considered. It has also been argued by Wu et al. (2018) that modeled $v_d$ is largely dependent on
prescribed minimum stomatal resistance ($r_{smin}$), and that uncertainties in $r_{smin}$ dominate simulation errors
in stomatal $O_3$ uptake. Here we found that the inclusion of LAI in light response function can largely
affect modeled stomatal conductance, leading to discrepancies in $v_d$. Despite that modifying prescribed
$r_{smin}$ can mitigate overall biases on a seasonal timescale, W89 still lacks the capabilities of simulating
the diurnal variation of stomatal $O_3$ uptake.

            All in all, we found that stomatal parameterization can significantly affect $v_d$ simulations. The
dry deposition schemes in current CTMs are parameterized in order to capture the average $O_3$ sink over
days or weeks, with less emphasis on smaller timescales such as diurnal cycles. In previous modeling
works, simulated biases in $v_d$ were usually attributed to uncertainties in LAI input or coarse model
resolution (Hardacre et al., 2015; Silva and Heald, 2018; Wu et al., 2018). In this study we emphasize
the importance of appropriately representing diurnal $v_d$ and $G_s$ variations in atmospheric modeling.
Diurnal $G_s$ variations and the late afternoon drop of $G_s$ caused by the temporal lag of VPD with PAR
and temperature have also been discussed in previous studies (Matheny et al., 2014; Zhang et al., 2014).
W89 uses a simplified stomatal representation that is highly dependent on the variation of solar
radiation, and thus simulated $G_s$ peaks with strongest sunlight despite that observed diurnal $G_s$ double-
peaks (the "boa" shape) when both water availability and sunlight are optimal. We found an overall
overestimation of $G_s$ by W89 especially for deciduous forest during the afternoon, which was also seen
by Lei et al. (2020), resulting in positive biases in simulated $v_d$. Z03 can better capture the observed
average $v_d$ diurnal cycles than W89 mainly due to the consideration of stomatal response to VPD.
Replacing FBB stomatal parameterization in W89 can reduce biases in simulated $v_d$ cycles. In general,
accounting for stomatal response to VPD and/or water stress using multiplicative or photosynthesis-
based stomatal algorithms can improve model performance in capturing diurnal variations of $G_s$ and $v_d$.

**3.3 Comparison of stomatal conductance schemes**

            Stomatal uptake dominates total ozone deposition during summer for long-term measurements
(Fig. S3). Accurate parameterization of stomatal resistance is important for not only seasonal, but also
diurnal courses of $v_d$ simulations. To investigate the capabilities of the four stomatal approaches, i.e.,





Eqs. (2) to (5), in capturing the spatial variations of $G_s$ across four major PFTs on a global scale, we
simulated $G_s$ at 68 FLUXNET sites using different stomatal algorithms. Since no direct observations of
canopy stomatal conductance or stomatal $O_3$ flux are available, here we used SynFlux $G_s$ derived from
$H_2O$ EC fluxes with the inverted Penman-Monteith equation to evaluate different stomatal approaches
(Ducker et al., 2018).

Figure 5 shows the comparison of daytime $G_s$ during growing periods using different stomatal
approaches for four major PFTs (PFT for tropical rainforest is not presented in SynFlux due to the
availability of corresponding $O_3$ measurements). The four stomatal approaches examined here can
generally capture the magnitudes of $G_s$ during the measuring periods. The multiplicative stomatal
approach in Z03 simulates $G_s$ with relatively low biases (NMBF = –0.07; NMAEF = 0.41) compared
with W89 which simulates high positive biases (NMBF = 0.25; NMAEF = 0.52). Z03 and FBB produce
similar biases, lower than MED or W89 in general. MED simulates $G_s$ with higher $R$-squared value ($R^2$
= 0.29) than other stomatal approaches ($R^2 \leq 0.18$). Statistic summary of monthly daytime $G_s$ for each
PFT is presented in Table 4. Different stomatal schemes simulate daytime $G_s$ within ± one standard
deviation evaluated using P-M $G_s$ for major PFTs. For deciduous broadleaf forests, W89 simulates
daytime $G_s$ with the highest positive mean biases (NMBF = 1.03), while Z03 has relatively low biases
(NMBF = 0.08). For the two photosynthesis-based stomatal approaches, FBB produces lower mean
biases (NMBF = 0.11) than MED (NMBF = 0.67). For evergreen needleleaf forests and crops, the four
stomatal algorithms can well reproduce P-M $G_s$, with |NMBF| < 0.07 and |NMBF| < 0.18, respectively.
For grasses, Z03 and FBB underestimate $G_s$ (NMBF = –0.43), while MED overestimates $G_s$ (NMBF =
0.44), and W89 simulates with NMAEF = 0.41 than other schemes (NMAEF > 0.50).





**Figure 5. Simulated and SynFlux daytime average canopy stomatal conductance ($G_s$) during growing seasons. Each point indicates daytime $G_s$ averaged over the growing seasons for the major PFT at one FLUXNET site.**

**Table 4. Statistic summary of monthly average daytime canopy stomatal conductance with two standard deviations (cm s$^{-1}$). DBF: Deciduous Broadleaf Forest; ENF: Evergreen Needleleaf Forest; CRO: Crop; GRA: Grass.**

| | | P-M | W89 | Z03 | FBB | MED | FBB (MAP $g_{1B}$) | MED (MAP $g_{1M}$) |
|---|---|---|---|---|---|---|---|---|
| **DBF** | mean±sd | 0.37±0.18 | 0.72±0.42 | 0.40±0.20 | 0.39±0.24 | 0.61±0.37 | 0.37±0.22 | 0.37±0.24 |
| | NMBF | / | 1.08 | 0.08 | 0.11 | 0.67 | 0.08 | 0.03 |
| | NMAEF | / | 1.08 | 0.28 | 0.32 | 0.69 | 0.27 | 0.27 |
| **ENF** | mean±sd | 0.29±0.13 | 0.25±0.17 | 0.24±0.13 | 0.25±0.19 | 0.25±0.19 | 0.30±0.23 | 0.31±0.26 |
| | NMBF | / | −0.01 | −0.07 | 0.03 | 0.00 | 0.11 | 0.15 |
| | NMAEF | / | 0.31 | 0.24 | 0.33 | 0.27 | 0.35 | 0.44 |
| **CRO** | mean±sd | 0.46±0.28 | 0.53±0.31 | 0.60±0.39 | 0.48±0.35 | 0.59±0.41 | 0.47±0.32 | 0.50±0.35 |
| | NMBF | / | 0.07 | 0.03 | −0.05 | 0.18 | -0.03 | 0.03 |





| | | | | | | | |
|---|---|---|---|---|---|---|---|
| | NMAEF | / | 0.40 | 0.41 | 0.45 | 0.47 | 0.40 | 0.44 |
| **GRA** | mean±sd | 0.43±0.29 | 0.37±0.24 | 0.25±0.18 | 0.26±0.20 | 0.57±0.46 | 0.29±0.25 | 0.32±0.29 |
| | NMBF | / | 0.00 | −0.43 | −0.43 | 0.44 | -0.39 | -0.27 |
| | NMAEF | / | 0.41 | 0.81 | 0.65 | 0.50 | 0.50 | 0.39 |

Overestimated $G_s$ using MED indicates systematic biases that can be associated with the prescribed slope parameters $g_{1M}$. The predictive strengths of FBB and MED are proved to be equal in previous studies when prescribed slope parameters $g_{1M}$ (Eq. 5) and $g_{1B}$ (Eq. 4) were fitted to leaf gas exchange measurements of dominant tree species (Franks et al., 2017; Franks et al., 2018). $g_{1M}$ and $g_{1B}$ were inferred from leaf-scale gas exchange measurements that might have spatial and temporal sampling biases (Lin et al., 2015). These sampling biases were found to be reduced by inferring $g_{1B}$ and $g_{1M}$ on the canopy scale using long-term EC measurements (Knauer et al., 2018). Medlyn et al. (2017) also found that the $g_{1M}$ values estimated from leaf-scale and canopy-scale measurements are not consistent across PFTs, and that using $g_{1M}$ derived from leaf-scale data can induce biases in canopy-scale simulations. Franks et al. (2018) proposed an approach for estimating the slope parameters based on observed linear relationship between mean annual precipitation (MAP) and the slope parameters $g_{1M}$ and $g_{1B}$. Parameterizing $g_{1M}$ and $g_{1B}$ with global MAP data can overcome the limitation of lacking spatiotemporal variations in current leaf-scale measurements, but it needs further validation with global observations. We therefore also tested MAP-derived $g_{1B}$ and $g_{1M}$ with the fitted functions described in Franks et al. (2018).

Figure 6 shows the comparison between simulated $G_s$ using MAP-derived slope parameters and P-M derived $G_s$ from SynFlux. The overall biases in simulated $G_s$ are reduced using MAP derived $g_{1B}$ and $g_{1M}$ compared with that using PFT-specific $g_{1B}$ and $g_{1M}$. Figure 7 shows comparison of simulated average daytime $G_s$ using MAP-derived $g_{1B}$ and $g_{1M}$ grouped for major PFTs. The simulated $G_s$ values using MAP-derived $g_{1B}$ and $g_{1M}$ are also summarized in Table 4 to compare with those using PFT-specific parameters. Both FBB and MED using MAP-derived $g_{1B}$ and $g_{1M}$ can reproduce $G_s$ comparable with P-M derived $G_s$ across different PFTs. The positive biases in MED-simulated $G_s$ for DBF, CRO, and GRA are reduced by using MAP-derived $g_{1M}$. MED-simulated $G_s$ that uses MAP as predictors of regional mean $g_{1M}$ is in better agreement with P-M $G_s$ than that using PFT-specific $g_{1M}$ on leaf scale. In previous studies, FBB and MED had equal predictive strengths when parameterized with site-specific leaf-scale data (Franks et al., 2018; Knauer et al., 2015). Our results also show that FBB and MED have comparable predictive strength when using MAP-derived $g_{1B}$ and $g_{1M}$.



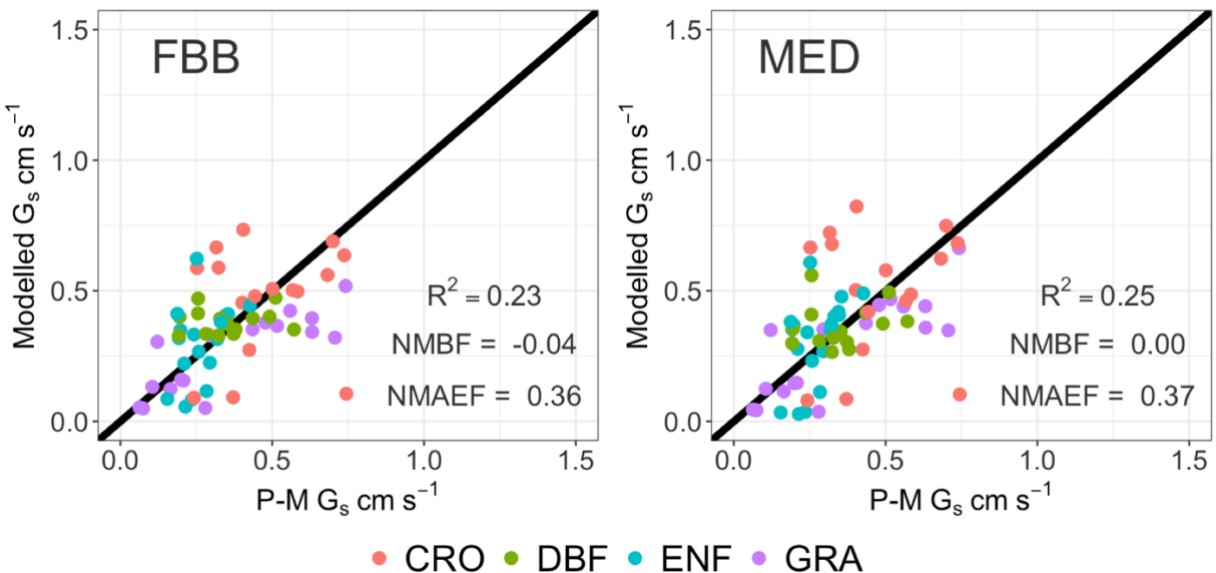

**Figure 6. FBB and MED using $g_{1B}$ and $g_{1M}$ derived from mean annual precipitation data compared with SynFlux canopy stomatal conductance ($G_s$) during growing seasons. Each point indicates average daytime $G_s$ for the major PFT at an individual FLUXNET site.**

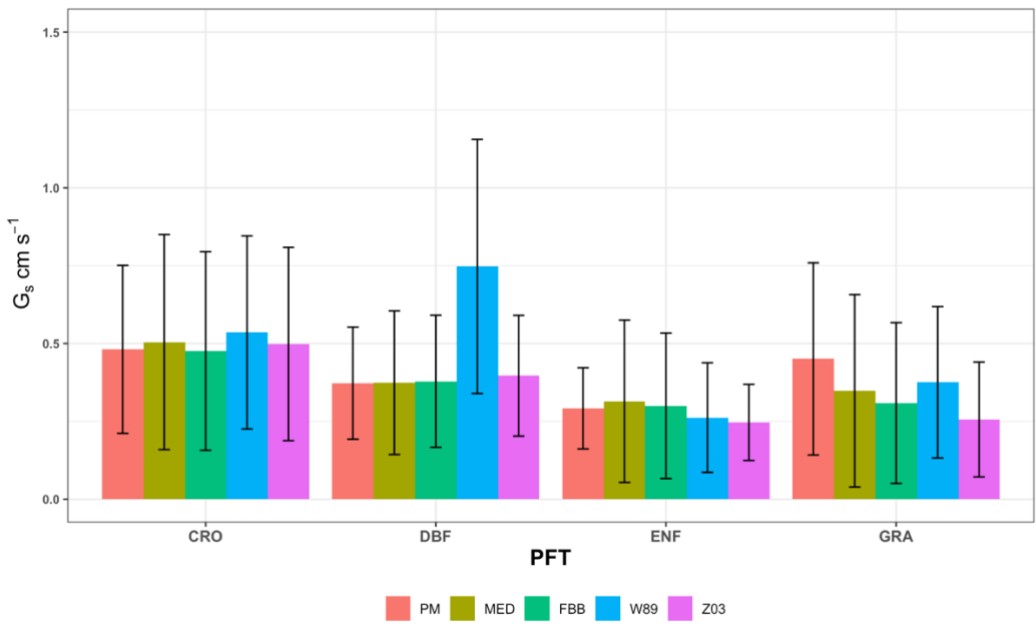

**Figure 7. Average daytime canopy stomatal conductance ($G_s$) computed with different stomatal conductance approaches for the four major PFTs. The error bars indicate two standard deviations. DBF: Deciduous Broadleaf Forest; ENF: Evergreen Needleleaf Forest; CRO: Crop; GRA: Grass.**





In general, Jarvis-type multiplicative and photosynthesis-based stomatal approaches have
comparable capabilities in reproducing the average inferred $G_s$ from SynFlux for major vegetation
types. The Jarvis-type stomatal parameterization in Z03 produces similar biases in $G_s$ as that using FBB
as shown in Table 4. MED produces higher $G_s$ values than FBB with PFT-specific slope parameters in
most cases. When using MAP-derived slope parameters, FBB and MED have similar predictive
strengths. The simplified stomatal approach in W89 is unable to capture the diurnal $G_s$ variations well
without the stomatal response to water stress, and that systematic positive biases in $G_s$ are found using
W89 especially for deciduous forests. The overestimated daytime $G_s$ simulated with W89 for deciduous
forest during growing seasons (Fig. 7) are also consistent with the overestimated daytime $v_d$ for
deciduous forest in JJA as shown in Table 3. Therefore, the positive biases in daytime $v_d$ for deciduous
forest are likely to be caused by the simplified representation of stomatal resistance in W89.

**3.4 Global simulations of $v_d$ and $G_s$**

We compared the global distribution of daytime $v_d$ and $G_s$ simulated with the six dry deposition
schemes. Simulated average July daytime $v_d$ and $G_s$ for year 2010 to 2014 with different model
configurations were compared under different $CO_2$ levels. Ambient $CO_2$ concentrations at 390 ppm
represents current $CO_2$ level. For all global simulations in this section, we used MERRA-2 meteorology
and MODIS LAI with a spatial resolution of 2°×2.5°. Simulated daytime $v_d$ and $G_s$ for each grid were
summed up by PFT fractions over vegetated land. Here we focus on the Northern Hemisphere where
high surface $O_3$ concentrations are typically observed during July.

Figure 8 shows July mean daytime $v_d$ during 2010–2014 over vegetated regions simulated with
the six dry deposition schemes. Z03 simulates lower daytime $v_d$ than W89 in most regions, except for
evergreen needleleaf regions at high latitudes where Z03 produces higher non-stomatal deposition rates
than W89 (Fig. 9a). W89FBB produces higher daytime $v_d$ for evergreen needleleaf regions, but lower
daytime $v_d$ for deciduous broadleaf regions compared with W89 (Fig. 9b). Our evaluation results in
Sect. 3.1 show that W89 overestimates observed daytime $v_d$ for deciduous broadleaf forests, but
underestimates $v_d$ for evergreen needleleaf forests (Fig. 1). W89FBB and Z03 can potentially better
capture observed $v_d$ than W89 in global simulations especially for evergreen needleleaf and deciduous
broadleaf regions.





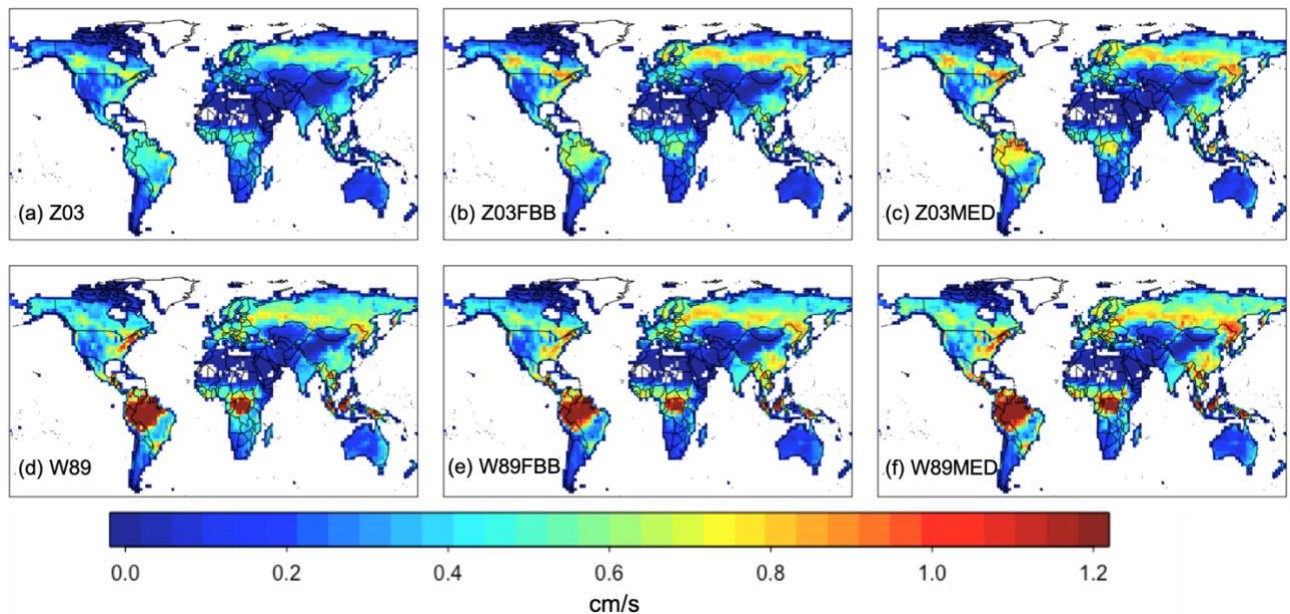

**Figure 8.** 2010–2014 July average daytime $v_d$ under 390 ppm $CO_2$ level simulated with the six dry deposition schemes: (a) Z03, (b) Z03FBB, (c) Z03MED, (d) W89, (e) W89FBB, (f) W89MED.


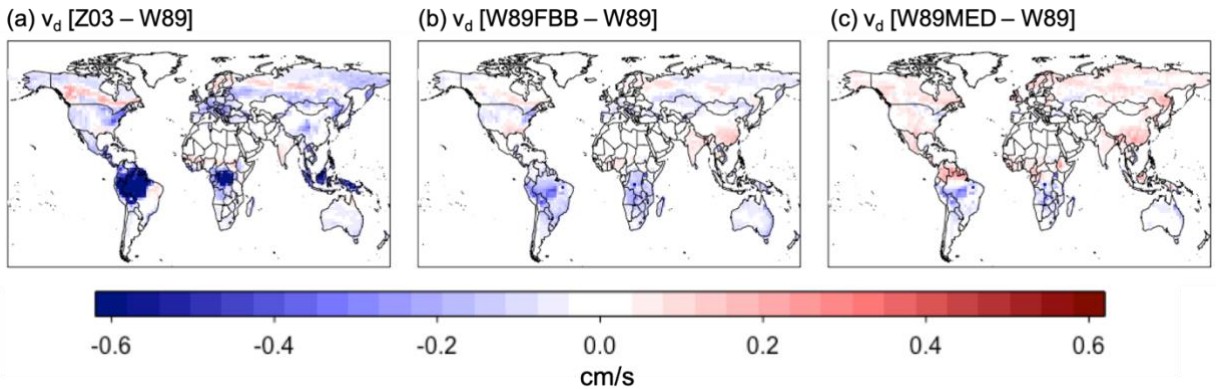

**Figure 9.** Differences in average daytime $v_d$ between different dry deposition schemes for 2010-2014 July.

Figure 10 shows simulated July daytime $G_s$ and the differences in simulated $G_s$ between
different stomatal approaches. Z03 simulates lower $G_s$ than W89 in general, except for some tropical
regions dominated with C4 grasses as well as some C3 crop regions in the Northern Hemisphere (Fig.
10e). Z03 also produces lower $G_s$ than FBB, except for some regions dominated with C4 grasses (Fig.
10g). Z03 produces lowest $G_s$ for evergreen broadleaf forests in the tropical regions and potentially
underestimates $G_s$ in these regions, which is also found by Wong et al. (2019). The multiplicative
stomatal parameterization in Z03 simulates lowest $G_s$ values compared with FBB, MED and W89 for





most regions. Z03 is developed for a regional air quality model focusing on North America and especially Canada, and has not been evaluated with tropical forest observations, leading to potential biases for tropical regions. The slope parameters $g_{1B}$ and $g_{1M}$ in FBB and MED for C4 species are lower than those for C3 species according to the higher water use efficiency of C4 species. However, C3 and
C4 photosynthesis pathways are not differentiated in Jarvis-type stomatal approaches, and this simplification in PFT classification can cause biases in $G_s$ simulated by W89 and Z03. MED produces higher $G_s$ than FBB (Fig. 10h) primarily due to the prescribed slope parameters as discussed in Sect. 3.3.

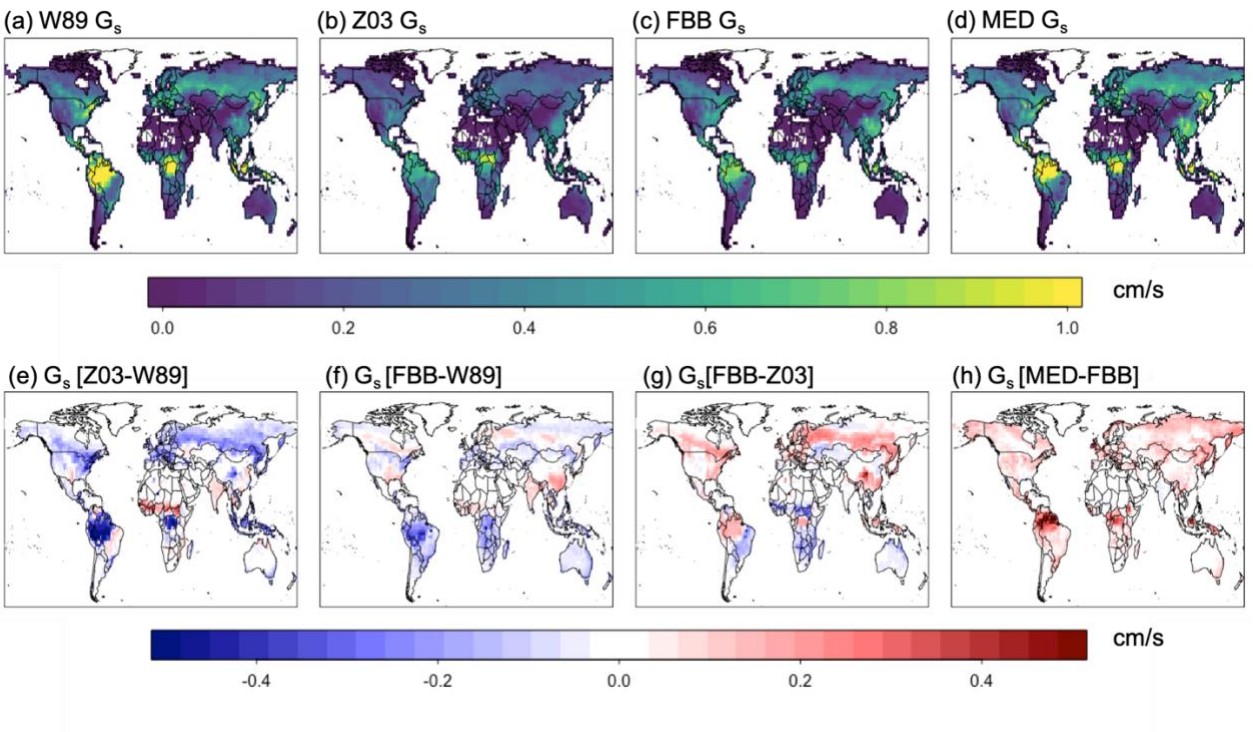


**Figure 10. Simulated average daytime $G_s$ with different stomatal schemes for 2010-2014 July.**

## 3.5 Sensitivity of stomatal conductance parameterization to elevated $CO_2$

To test the changes in $O_3$ dry deposition velocity due to stomatal conductance closure alone
under rising $CO_2$ levels, we conducted simulations with only variations in the choice of stomatal algorithms. Prescribed present-day meteorology and land use were applied for all simulations. Differences in simulated $G_s$ between photosynthesis-based and multiplicative stomatal parameterization were compared. We also conducted experiments with ambient $CO_2$ concentrations at 550 ppm and 1370 ppm, which represent future $CO_2$ levels under RCP8.5 scenarios in 2050 and 2100 respectively. The
stomatal approaches used in current LSMs are developed for short-term stomatal responses, and are





assumed to be adequate for long-term responses by accounting for the $CO_2$ effect on stomatal conductance via the FBB model. Jarvis-type multiplicative schemes do not generally represent any ecophysiological responses to rising $CO_2$, so we added an empirical $CO_2$ response function derived from photosynthesis-stomatal conductance model. Franks et al. (2013) summarized and tested a

generalized formulation for long-term net $CO_2$ assimilation rate ($A_n$) vs atmospheric $CO_2$ concentration ($c_a$) is as follows:

$$A_{n(rel)} \approx \left[\frac{(c_a - \Gamma^*)(c_{a0} + 2\Gamma^*)}{(c_a + 2\Gamma^*)(c_{a0} - \Gamma^*)}\right], \tag{7}$$

where $A_{n(rel)}$ is the relative change in $A_n$, $c_{a0}$ is the reference atmospheric $CO_2$ concentration, $\Gamma^*$ is the $CO_2$ compensation point without dark respiration. This expression for $A_{n(rel)}$ is based on the assumption

of optimized RuBP regeneration-limited photosynthesis in a nitrogen-limited system. The relative change in stomatal conductance is accordingly described as:

$$g_{w(rel)} \approx \frac{A_{n(rel)}}{c_{a(rel)}}, \tag{8}$$

where $g_{w(rel)}$ and $c_{a(rel)}$ are leaf stomatal conductance and atmospheric $CO_2$ concentration, respectively, relative to the value in a similar system at constant current ambient $CO_2$ concentration. We therefore

applied Eq. (7) and Eq. (8), multiplying Eq. (8) to $G_s$, to represent stomatal response to $CO_2$ changes in the Jarvis-type approaches. Here we focus on the differences in simulated stomatal response to $CO_2$ levels alone on a global scale between photosynthesis-based and Jarvis-type stomatal parameterization.

Figure 11 shows the changes of simulated $G_s$ using multiplicative and photosynthesis-based stomatal approaches under different $CO_2$ levels. Comparison of simulated $G_s$ between 550 ppm and 390

ppm $CO_2$ levels is shown in the left panel of Figure 11. The average July daytime $G_s$ values under 550 ppm $CO_2$ simulated with FBB and MED are reduced 14% and 19% respectively compared with current $CO_2$ level, lower than the relative change of –35% using Jarvis-type scheme with empirical response function (Fig. 11e). Comparison of simulated $G_s$ between 1370 ppm and 390 ppm $CO_2$ levels is shown in the right panel of Figure 11. FBB and MED simulate –46% and –58% reduction respectively in

average daytime $G_s$, while using the Jarvis-type scheme with empirical response function gives –77% reduction in average daytime $G_s$. The global average $G_s$ computed with FBB and MED are generally less sensitive to $CO_2$ changes than the empirical long-term response function. Simulated $G_s$ with MED is more sensitive to elevated $CO_2$ concentrations than FBB due to the prescribed $g_{1M}$ values. The long-term forest tree Free-Air $CO_2$ Enrichment (FREE) experiments have found reductions in $G_s$ of ~20% on

average under 550 ppm $CO_2$ level (Ainsworth and Long, 2005), which is more consistent with what we found using the photosynthesis-based schemes than the multiplicative scheme. Yet, the magnitude of reductions in $G_s$ varies across studies, ranging about 10–39% (Herrick et al., 2004; Tricker et al., 2009; Warren et al., 2011). Using the empirical $CO_2$ response function in Jarvis-type stomatal approaches gives a more sensitive response to elevated $CO_2$ levels than photosynthesis-based approaches. Previous

studies have found that terrestrial biosphere models using photosynthesis-based stomatal approaches combined with mechanistic parameterization of nitrogen limitation can better reproduce observed responses to $CO_2$ enrichment experiments (Lawrence et al., 2019; Wieder et al., 2019). The Jarvis-type multiplicative stomatal approach without more mechanistic representation of complex ecophysiological constraints (e.g., nitrogen limitation, ozone damage) would likely exaggerate stomatal closure effects

with higher simulated reductions in $G_s$ under rising $CO_2$ levels in future predictions.





**Figure 11. Changes in July daytime average $G_s$ simulated with FBB, MED and W89 (using empirical $CO_2$ response function) under different $CO_2$ levels.**

## 4 Conclusions and discussion

This study provides an intercomparison and evaluation of dry deposition schemes, with highlights on the choice of stomatal parameterization and the importance of representing ecophysiological processes in atmospheric models. Different dry deposition and stomatal conductance schemes were implemented in a terrestrial biosphere model driven by consistent prescribed meteorological fields and land cover data to isolate the impacts of choices of model parameterization. We evaluated the most widely used dry deposition schemes against globally distributed observations. We also compared and evaluated the state-of-art photosynthesis-based stomatal conductance algorithms using FLUXNET measurements. Our analysis shows the importance of advancing the treatment of





stomatal conductance in dry deposition schemes within current CTMs, which is essential for modeling $O_3$ air quality under climate change, especially in relation to plant responses to water stress.

All the tested dry deposition schemes in this study can generally capture the observed seasonal average $v_d$ for major PFTs. Multiplicative W89 and Z03 reproduce observed seasonal $v_d$ with similar mean and absolute biases. Z03FBB, consisting of the photosynthesis-based FBB stomatal approach and Z03 non-stomatal parameterization, generally performs better in capturing observed seasonal daytime

$v_d$. Z03 can better simulate diurnal $v_d$ variations than W89, and can also capture observed $G_s$ with similar mean biases as FBB for major PFTs on different timescales. W89 was parameterized to capture average $v_d$ over weeks in the early generation of CTMs, and was guaranteed to reproduce seasonal observations well. Therefore, the stomatal resistance in W89 was parameterized rather simply to simulate the magnitude of observed stomatal resistance averaged over weeks accordingly (Wesely,

1989). The major difference between Z03 and W89 in the stomatal resistance calculation is whether a VPD response function is included. The misrepresentation of diurnal $v_d$ variations due to lack of water stress response in W89 can potentially cause higher biases in simulated $O_3$ sink since the covariation of surface $O_3$ and stomatal conductance is based on an hourly or even half hourly timescale. The Wesely scheme in current CTMs should urgently be revised for present-day simulations to better capture diurnal

variations and plant responses to water stress, which was also recommended in previous studies (Emmerichs et al., 2020; Lin et al., 2019; Niyogi et al., 1998). Despite that adding a biospheric module with photosynthesis-based stomatal schemes may have additional computational cost (Lei et al., 2020), having a photosynthesis-based stomatal scheme or fully coupling dry deposition simulation in CTMs with a biosphere model would be the preferred approach for projecting future $O_3$ air quality under

changing $CO_2$ concentration and climate. The non-stomatal parameterization in both W89 and Z03 should also be updated to better reflect our current understanding of non-stomatal sinks (Clifton et al., 2020).

The MED scheme based on the optimization stomatal theory with PFT-specific slope parameters from Lin et al. (2015) may overestimate $G_s$. We found that using the revised slope parameters may

mitigate the high biases in simulated $G_s$, indicating the potential of using the slope parameters derived from global precipitation data. Current climate models lack the capability to predict hydroclimate variabilities accurately, making it difficult to link precipitation with the slope parameters in model simulations especially when precipitation is expected to be changing under climate change. Using PFT-specific slope parameters derived from globally distributed leaf-level measurements can also better

capture the features of different plant species than using generic categories of C3 and C4 photosynthetic pathways. Gaps remain in understanding the spatiotemporal variations of the slope parameters in FBB and MED despite their critical role as the indicators of the intrinsic plant water use efficiency, regulated by species-related characteristics and environmental factors (Manzoni et al., 2011; Miner et al. 2017).

Disagreement was found in the spatial distribution of simulated $v_d$ and $G_s$ using different dry

deposition schemes, similar to that found previously by Wong et al. (2019). Differences in both stomatal and non-stomatal parameterization cause regional disagreement especially for tropical forests. Comparing to the SynFlux-inferred $G_s$, we found potential overestimation of $G_s$ for deciduous broadleaf forests by W89 and underestimation of $G_s$ for evergreen needleleaf forests by Z03 on a global scale. As the inference of canopy-scale $G_s$ can be improved by advances in partitioning transpiration and

evaporation (Stoy et al., 2019), using ecosystem-scale measurements (e.g., FLUXNET) to calibrate



stomatal schemes can help to overcome the limitation of leaf-level measurements in spatiotemporal coverage.

The impacts of increasing atmospheric $CO_2$ on the terrestrial carbon sink is of great importance for land surface and climate modeling (Fatichi et al., 2019; Wieder et al., 2019). However, large

uncertainties remain in the prediction of stomatal responses to climate change. The short-term variability in simulated leaf-level stomatal conductance under elevated $CO_2$ levels mainly depend on meteorological conditions, while model parameters are more dominant in longer timescales, and thus stomatal conductance parameterization is of great importance in determining land-atmosphere interactions under future scenarios (Paschalis et al., 2017). Multiplicative and photosynthesis-based

stomatal schemes simulate different sensitivities of stomatal conductance to rising $CO_2$ concentrations. Our attempt to include the empirical $CO_2$ response function of Franks et al. (2013) in multiplicative stomatal schemes result in a much larger reduction in global $G_s$ that doubled the average relative change computed with photosynthesis-based stomatal schemes, and potentially overstates stomatal responses to elevated $CO_2$ under future scenarios.

In general, for atmospheric model development endeavoring to better simulate biosphere-atmosphere fluxes relevant for atmospheric chemistry, accounting for plant photosynthetic processes and other ecophysiological responses to varying environmental conditions is important especially for future predictions under changing climate and atmospheric composition. For present-day simulations of dry deposition, despite the overall performance of different deposition schemes being similar, PFT-

specific or region-specific projections have large discrepancies due to different stomatal and non-stomatal parameterization. Long-term field measurements that provide hourly flux observations for major vegetation types will benefit not only stomatal and non-stomatal parameterization from diurnal to seasonal timescales, but also ecophysiological representation in atmospheric models at large, with potential to improve modeled air quality forecasts.


**Acknowledgements**

This work was supported by the Research Grants Council (RGC) General Research Fund (GRF; Proj. No.: 14306220) awarded to A. P. K. Tai.

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
