# Peer review of "Influence of plant ecophysiology on ozone dry deposition: Comparing between multiplicative and photosynthesis-based dry deposition schemes and their responses to rising CO2 level"

_Biogeosciences, 2021_

## Author Comment (AC1)

**Response to Reviewers' Comments on "Influence of plant ecophysiology on ozone dry deposition: Comparing between multiplicative and photosynthesis-based dry deposition schemes and their responses to rising CO₂ level" by Sun et al.**

The comments of the referee are given as plain text, while the authors' response is given in *italic.* We have revised the paper based on the reviewers' comments.

***Response to Referee #1***
In this paper, the authors use a standalone terrestrial biosphere model to evaluate both multiplicative and photosynthesis-based schemes of stomatal conductance of ozone. Observational datasets of the dry deposition velocity and the stomatal conductance of ozone are used to do the model evaluation. The authors suggested that the photosynthesis-based stomatal algorithms that captured the responses to water stress had a better agreement with the observations. The manuscript describes a straightforward modeling study exploring basic parameterizations and comparisons to observations, and fits into the scope of Biogeosciences. I have a few minor comments as listed below.

- *We thank the referee for the very helpful comments and suggestions. The paper has been revised accordingly. Our point-by-point responses are provided below.*

My major concern is that based on the model-observation comparison in this paper, I do not see a significant improvement by using photosynthesis-based stomatal conductance methods, compared to the traditional multiplicative methods. The default multiplicative W89 scheme without stomatal response to water stress fails to reproduce the diurnal variations in Gs, but the multiplicative Z03 method seems to agree well with the other photosynthesis-based methods and the observations. Furthermore, all schemes compare poorly with observations in rainforests and in the Blodgett forest site (which is often associated with higher temperatures and water stress). Can the authors comment a bit more on the advantages of using photosynthesis-based methods?

- *We very much agree that using photosynthesis-based stomatal conductance models do not significantly improve model performance over Z03, and that Z03 agrees well with photosynthesis-based methods. In the paper, we have therefore mentioned this at various places, highlighting that both photosynthesis-based schemes and Z03 multiplicative schemes are better than W89, mostly likely due to their ability to capture plant responses to water stress and VPD, e.g., P15 L442: "… In general, accounting for stomatal response to VPD and/or water*

*stress using multiplicative or photosynthesis-based stomatal algorithms can improve model performance in capturing diurnal variations of $G_s$ and $v_d$.”*

- *The merits of multiplicative methods such as lower computational costs and higher compatibility are undeniable for Earth system modeling. Multiplicative methods parameterized with observations can also be improved whenever more field measurements are available. Yet, with more biophysically meaningful and measurable properties, photosynthesis-based methods are principally more mechanistic than multiplicative stomatal methods, and can better address plant responses to a changing environment (e.g., rising $CO_2$ and temperature) with rapidly expanding knowledge from biologists. Furthermore, parameters for photosynthesis-based methods can be obtained from leaf-scale measurements, which overall cost less than dry deposition flux measurements that are used for parameterizing multiplicative model. We have now emphasized these points more in the Conclusions and Discussion, e.g., P26 L681: “Our attempt to include the empirical $CO_2$ response function of Franks et al. (2013) in multiplicative stomatal schemes result in a much larger reduction in global $G_s$ that doubled the average relative change computed with photosynthesis-based stomatal schemes, and potentially overstates stomatal responses to elevated $CO_2$ under future scenarios.”*

The numbers and names of the modeling schemes are sometimes confusing. For example, the words “multiplicative” and “photosynthesis-based” in the title refer to stomatal conductance schemes, not dry deposition schemes, right? In the abstract, the Medlyn scheme is also a photosynthesis-based method, so there are actually two multiplicative (W89, Z03), two photosynthesis-based (FBB, MED) stomatal conductance schemes. I think it should be stated clearly in the abstract and introduction, or it will confuse the readers.

- *Thanks for your suggestions. Yes, “multiplicative” and “photosynthesis-based” refer to stomatal conductance schemes. The relevant parts are revised accordingly.*

  *“We developed and used a standalone terrestrial biosphere model, driven by a unified set of prescribed meteorology, to evaluate two widely used dry deposition modeling frameworks, Wesely (1989) and Zhang et al. (2003), with different configurations of stomatal resistance: 1) the default multiplicative method in the Wesely scheme (W89) and Zhang et al. (2003) scheme (Z03); 2) the traditional photosynthesis-based Farquhar-Ball-Berry (FBB) stomatal algorithm; 3) the Medlyn stomatal algorithm (MED) based on optimization theory.”*

Also, the figures should be consistent to show all 6 schemes when comparing to observed dry deposition velocity Vd, and show all 4 schemes when comparing to observed stomatal conductance Gs. For example, why not compare the Z03 scheme in Figure 11?

- *Thank you for your suggestion. We updated the Figure 11 to make it consistent.*

[Figure]

L42: Does this "45%" refer to an annually averaged percentage? How does this compare to your results? As the stomatal conductance is the main focus of this paper, I would suggest moving Figure S3 (showing the fraction of stomatal conductance to total deposition) to the main text.

- *Thanks for the suggestion. Yes "45%" refers to annual daytime average (Clifton et al., 2020), which is stomatal fraction of ozone dry deposition aggregated from previous literature. We aggregate annual daytime stomatal fraction here using SynFlux: W89 (87%), Z03 (62%), FBB (65%), MED (68%). Stomatal fractions in Clifton et al. (2020) are calculated with P-M method. Our results show that Z03 and FBB agree with P-M derived stomatal conductance. Simulated higher stomatal fractions can be related with underestimation of non-stomatal conductance. Not all datasets in Clifton et al. include data from all seasons. The*

*magnitude of stomatal fraction is also affected by vegetation types: deciduous forest has higher stomatal fraction than other vegetation types. We have now revised the relevant parts and moved Figure S3 to main text as suggested.*

L257: Please briefly explain the P-M method here.

- *We have now explained the P-M method in the Supplementary.*

  *"We use evaporative-resistance form of Penman-Monteith method to keep consistent with SynFlux stomatal conductance. The leaf stomatal conductance is:*

  $$g_w^{-1} = \frac{\varepsilon \rho (e_s(T_f) - e)}{pE} - (r_a + r_{b,w}),$$

  *where ε is mass ratio between water and dry air, p is air pressure, E is surface moisture flux, $T_f$ is leaf temperature, $e_s(T_f)$ is the saturation vapor pressure at leaf surface. $r_a$ is aerodynamic resistance, $r_{b,w}$ is quasi-laminar layer resistance to water vapor. $T_f$ is estimated as follows:*

  $$T_f = T + \frac{H(r_a + r_{b,H})}{c_p \rho},$$

  *where T is air temperature, H is sensitive heat, $c_p$ is specific heat of air, ρ is the mass density of air, $r_{b,H}$ is quasi-laminar layer resistance to heat.*

  *Stomatal conductance of $O_3$ is calculated with molecular diffusion coefficient ratio 0.6 between $O_3$ and water vapor:*

  $$g_s = 0.6 g_w \text{ "}$$

Table 3: This table contains a lot information and is not easy to read. How about using some background colors, e.g., red/blue to show overestimation/underestimation and dark/light colors to indicate large/small bias?

- *Thank you for the suggestion. Updated Table 3 with background colors.*

| PFT | Season | Observation | | W89 | | | W89FBB | | | W89MED | | | Z03 | | | Z03FBB | | | Z03MED | | |
|---|---|---|---|---|---|---|---|---|---|---|---|---|---|---|---|---|---|---|---|---|---|
| | | mean±sd | | mean±sd | NMBF | NMAEF | mean±sd | NMBF | NMAEF | mean±sd | NMBF | NMAEF | mean±sd | NMBF | NMAEF | mean±sd | NMBF | NMAEF | mean±sd | NMBF | NMAEF |
| DBF | JJA | 0.69±0.10 | | 0.90±0.17 | 0.32 | 0.32 | 0.59±0.10 | −0.16 | 0.26 | 0.81±0.24 | 0.18 | 0.41 | 0.55±0.09 | −0.25 | 0.30 | 0.58±0.11 | −0.18 | 0.26 | 0.78±0.25 | 0.14 | 0.41 |
| | MAM | 0.33±0.02 | | 0.42±0.13 | 0.27 | 0.43 | 0.28±0.08 | −0.21 | 0.23 | 0.35±0.10 | 0.05 | 0.26 | 0.29±0.08 | −0.13 | 0.27 | 0.31±0.05 | −0.09 | 0.18 | 0.37±0.08 | 0.10 | 0.21 |
| | SON | 0.52±0.20 | | 0.49±0.12 | −0.05 | 0.18 | 0.29±0.07 | −0.78 | 0.78 | 0.39±0.13 | −0.34 | 0.34 | 0.41±0.06 | −0.26 | 0.26 | 0.37±0.05 | −0.39 | 0.39 | 0.46±0.11 | −0.11 | 0.13 |
| | DJF | 0.25±0.08 | | 0.14±0.05 | −0.86 | 0.97 | 0.14±0.05 | −0.86 | 0.86 | 0.15±0.06 | −0.72 | 0.87 | 0.24±0.04 | −0.04 | 0.21 | 0.26±0.03 | 0.02 | 0.23 | 0.26±0.04 | 0.05 | 0.27 |
| ENF | JJA | 0.58±0.23 | | 0.46±0.12 | −0.29 | 0.35 | 0.46±0.11 | −0.30 | 0.42 | 0.47±0.10 | −0.27 | 0.40 | 0.42±0.14 | −0.39 | 0.68 | 0.52±0.14 | −0.14 | 0.44 | 0.53±0.13 | −0.12 | 0.42 |
| | MAM | 0.46±0.15 | | 0.35±0.11 | −0.31 | 0.43 | 0.34±0.10 | −0.34 | 0.40 | 0.37±0.12 | −0.24 | 0.37 | 0.42±0.06 | −0.10 | 0.31 | 0.43±0.09 | −0.07 | 0.26 | 0.46±0.11 | −0.01 | 0.26 |
| | SON | 0.47±0.22 | | 0.35±0.12 | −0.35 | 0.43 | 0.28±0.07 | −0.64 | 0.68 | 0.26±0.04 | −0.83 | 0.85 | 0.39±0.13 | −0.21 | 0.46 | 0.41±0.15 | −0.13 | 0.37 | 0.40±0.12 | −0.18 | 0.43 |
| | DJF | 0.32±0.21 | | 0.17±0.07 | −0.87 | 0.89 | 0.19±0.08 | −0.66 | 0.73 | 0.16±0.06 | −0.98 | 1.01 | 0.30±0.11 | −0.08 | 0.29 | 0.30±0.15 | −0.05 | 0.28 | 0.28±0.12 | −0.14 | 0.36 |
| CRO | / | 0.53±0.16 | | 0.50±0.26 | −0.05 | 0.29 | 0.72±0.15 | 0.37 | 0.43 | 0.81±0.13 | 0.54 | 0.54 | 0.54±0.11 | 0.03 | 0.18 | 0.61±0.15 | 0.16 | 0.32 | 0.67±0.14 | 0.27 | 0.32 |
| TRF | / | 0.76±0.48 | | 1.11±0.07 | 0.46 | 0.56 | 0.98±0.06 | 0.29 | 0.52 | 1.10±0.10 | 0.44 | 0.53 | 0.47±0.05 | −0.60 | 0.85 | 0.57±0.04 | −0.33 | 0.61 | 0.66±0.07 | −0.14 | 0.48 |
| GRA | JJA | 0.33±0.17 | | 0.72±010 | 1.21 | 1.21 | 0.59±0.21 | 0.82 | 0.82 | 0.84±0.28 | 1.56 | 1.56 | 0.50±0.12 | 0.53 | 0.79 | 0.50±0.16 | 0.51 | 0.51 | 0.68±0.21 | 1.08 | 1.08 |
| | MAM | 0.39±0.13 | | 0.58±0.13 | 0.48 | 0.48 | 0.43±0.00 | 0.08 | 0.28 | 0.62±0.16 | 0.57 | 0.74 | 0.42±0.11 | 0.06 | 0.48 | 0.46±0.03 | 0.17 | 0.36 | 0.62±0.15 | 0.56 | 0.72 |
| | SON | 0.30±0.06 | | 0.59±0.21 | 1.00 | 1.20 | 0.46±0.22 | 0.55 | 0.78 | 0.55±0.26 | 0.88 | 1.03 | 0.42±0.29 | 0.43 | 0.76 | 0.46±0.20 | 0.54 | 0.80 | 0.54±0.22 | 0.82 | 0.95 |
| | DJF | 0.33±0.05 | | 0.34±0.26 | 0.02 | 0.68 | 0.24±0.14 | −0.37 | 0.56 | 0.34±0.31 | 0.04 | 0.77 | 0.31±0.15 | −0.08 | 0.46 | 0.35±0.18 | 0.06 | 0.49 | 0.443±0.32 | 0.31 | 0.79 |

L327 Not sure what this sentence means. Do you mean ozone reacts "with" soil-emitted NO and BVOC here?

- *Yes. Revised as suggested:*

  *"Non-stomatal $O_3$ deposition includes chemical reactions of $O_3$ with nitric oxide (NO) and biogenic volatile organic compounds (BVOC) from soil emissions (Fares et al., 2012)."*

Figure 2 The models seem to predict an overall earlier peak than the observations. Can the authors comment on why it could be?

- *Simulated monthly daytime $v_d$ peaks in around June to July, while observed daytime $v_d$ peaks during July to August. As we used observed LAI in this study, LAI is not the major driver as in previous studies. For long-term sites in Figure 2, overestimation of stomatal conductance and underestimation of non-stomatal conductance cause discrepancies between modelled and observed $v_d$. Simulated and observed monthly daytime average stomatal conductance variations are shown in the figure below. The peaks in early summer are mainly driven by stomatal conductance, due to favorable conditions such as higher solar radiation and lower VPD.*

[Figure]

L401 which site is "ponderosa pine forest"? Include the site name here.

- *Blodgett Ameriflux site. Revised as suggested:*

  *"The major $O_3$ removal process in the ponderosa pine plantation at the Blodegett Ameriflux site is non-stomatal $O_3$ sink through in-canopy chemical reactions between $O_3$ and BVOC (Fares et al., 2010; Kurpius and Goldstein, 2003)."*

Finally, this manuscript includes many abbreviations and sometimes is hard to follow. I would suggest including a list of abbreviations and explanations if possible.

- *Thanks for your suggestion. Revised as suggested.*

Table. List of abbreviations used in this paper with descriptions.

| Symbol | Description |
|---|---|
| $A_n$ | leaf net $CO_2$ assimilation rate |
| BVOC | biogenic volatile organic compounds |
| CLM | Community Land Model |
| CRO | Crop |
| $C_s$ | $CO_2$ concentration at the leaf surface |
| CTMs | chemical transport models |
| DBF | Deciduous Broadleaf Forest |
| $D_i$ | molecular diffusivities for water |
| $DO_3SE$ | The Deposition of $O_3$ for Stomatal Exchange |
| $D_v$ | molecular diffusivities for pollutant gas |
| ENF | Evergreen Needleleaf Forest |
| ESMs | Earth system models |

| FBB | Farquhar-Ball-Berry stomatal scheme |
|---|---|
| $g_0$ | PFT-dependent minimum stomatal conductance |
| $g_{1B}$ | fitted slope parameter for Ball-Berry model |
| $g_{1M}$ | fitted slope parameter for Medlyn model |
| GRA | Grass |
| $G_s$ | Canopy stomatal conductance |
| $h_s$ | leaf surface relative humidity |
| $L$ | Obukhov length |
| LAI | leaf area index |
| $L^{sha}$ | shaded LAI |
| LSMs | land surface models |
| $L^{sun}$ | sunlit LAI |
| MAP | mean annual precipitation |
| MED | Medlyn stomatal scheme |
| MERRA-2 | Modern-Era Respective analysis for Research and Applications version 2 |
| MODIS | Moderate Resolution Imaging Spectroradiometer |
| NMAEF | normalized mean absolute error factor |
| NMBF | normalized mean bias factor |
| NO | nitric oxide |
| $O_3$ | ozone |
| P-M | Penman-Monteith |
| PAR | photosynthetically active radiation |
| PFTs | plant functional types |
| $P_r$ | the Prandtl number for air |
| $R^2$ | $R$-squared value |
| $R_a$ | aerodynamic resistance |
| $R_{ac}$ | in-canopy aerodynamic resistance |
| $R_{adc}$ | lower canopy aerodynamic resistance |
| $R_{ag}$ | ground aerodynamic resistance |
| $R_b$ | quasi-laminar sublayer resistance |
| $r_b$ | leaf boundary resistance |
| $R_c$ | bulk surface resistance |
| $R_c$ | canopy resistance |
| $R_{clx}$ | lower canopy resistance |
| $R_{cut}$ | cuticular resistance |
| $R_{cutd0}$ | reference cuticular resistance for dry condition |
| $R_{cutw0}$ | reference cuticular resistance for wet condition |
| $R_g$ | ground resistance |
| RH | relative humidity |
| $R_s$ | stomatal resistance |
| $r_{smin}$ | minimum stomatal resistance |
| $r_s^{sha}$ | shaded stomatal resistance |

| | |
|---|---|
| $r_s^{sun}$ | sunlit stomatal resistance |
| RuBP | ribulose 1,5-bisphosphate |
| $S_r$ | the Schmidt number |
| SRAD | incoming shortwave solar radiation |
| SW | soil wetness |
| $T$ | surface temperature |
| TEMIR | Terrestrial Ecosystem Model in R |
| TRF | Tropical Rainforest |
| $u_*$ | friction velocity |
| $v_d$ | dry deposition velocity of $O_3$ |
| VPD | vapor pressure deficit |
| W89 | Wesely deposition scheme |
| W89FBB | Wesely deposition scheme replaced with Faquhar-Ball-Berry stomatal scheme |
| W89MED | Wesely deposition scheme replaced with Medlyn stomatal scheme |
| $W_{st}$ | stomatal blocking factor |
| $z$ | reference height |
| $z_0$ | roughness height |
| Z03 | Zhang et al. (2003) deposition scheme |
| Z03FBB | Zhang et al. (2003) deposition scheme replaced with Faquhar-Ball-Berry stomatal scheme |
| Z03MED | Zhang et al. (2003) deposition scheme replaced with Medlyn stomatal scheme |
| $\kappa$ | von Kármán constant |
| $\psi$ | water stress |

---

## Author Comment (AC2)

**Response to Reviewers' Comments on "Influence of plant ecophysiology on ozone dry deposition: Comparing between multiplicative and photosynthesis-based dry deposition schemes and their responses to rising CO₂ level" by Sun et al.**

The comments of the referee are given as plain text, while the authors' response is given in *italic.* We have revised the paper based on the reviewers' comments.

**Response to Referee #2**

The authors examine the impact of carrying the stomatal conductance parameterization on the predicted deposition velocity. Comparisons are made between the well-used Wesely (1989) and Zhang et al. (2003) approaches as well as derivative parameterizations developed by substituting two different photosynthesis-based approaches into both the Wesely and Zhang approaches. Results are compared against observational data from several flux studies as well as the SynFlux dataset. Overall, the paper is sound but could benefit from additional editing. The text in Table 3 is very small as is the text in Figure 1. I am surprised that the photosynthetic models did not offer greater improvements in modeled values and would be interested in your thoughts on that. I would also be interested in seeing values of the correlation coefficient.

- *We thank the reviewer for the insightful comments and suggestions. The manuscript has been revised accordingly.*
- *We agree that photosynthesis-based stomatal conductance models do not significantly improve model performance. In the paper, we have therefore mentioned this at various places, highlighting that both photosynthesis-based schemes and Z03 multiplicative schemes are better than W89, mostly likely due to their ability to capture plant responses to water stress and VPD, e.g., P15 L442: "... In general, accounting for stomatal response to VPD and/or water stress using multiplicative or photosynthesis-based stomatal algorithms can improve model performance in capturing diurnal variations of $G_s$ and $v_d$."*
- *The merits of multiplicative methods such as lower computational costs and higher compatibility are undeniable for Earth system modeling. Multiplicative methods parameterized with observations can also be improved whenever more field measurements are available. Yet, with more biophysically meaningful and measurable properties, photosynthesis-based methods are principally more mechanistic than multiplicative stomatal methods, and can better address plant responses to a changing environment (e.g., rising $CO_2$ and temperature) with rapidly expanding knowledge from biologists. Furthermore, parameters for photosynthesis-based methods can be obtained from leaf-scale measurements, which overall cost less than dry deposition flux measurements that are used for parameterizing multiplicative model. We have now emphasized these points more in the Conclusions and Discussion, e.g., P26 L681: "Our attempt to include the empirical $CO_2$ response function of Franks et al. (2013) in multiplicative stomatal schemes result in a much larger reduction in global $G_s$ that doubled the average relative change computed with photosynthesis-based stomatal schemes, and potentially overstates stomatal responses to elevated $CO_2$ under future scenarios."*

  *We calculated Pearson correlation coefficient R for seasonal average modelled and observed $v_d$ at all observational sites: W89 (R = 0.61), W89FBB (R = 0.56), W89MED (R = 0.48), Z03 (R = 0.37), Z03FBB (R = 0.47), Z03MED (R = 0.41). The p-values for all coefficients here are <0.01.*

  *We updated font size of Table 3 and Figure 1 with colorblind friendly palette as below:*

[Figure]

| PFT | Season | Observation mean±sd | W89 mean±sd | NMBF | NMAEF | W89FBB mean±sd | NMBF | NMAEF | W89MED mean±sd | NMBF | NMAEF | Z03 mean±sd | NMBF | NMAEF | Z03FBB mean±sd | NMBF | NMAEF | Z03MED mean±sd | NMBF | NMAEF |
|---|---|---|---|---|---|---|---|---|---|---|---|---|---|---|---|---|---|---|---|---|
| DBF | JJA | 0.69±0.10 | 0.90±0.17 | 0.32 | 0.32 | 0.59±0.10 | -0.16 | 0.26 | 0.81±0.24 | 0.18 | 0.41 | 0.55±0.09 | -0.25 | 0.30 | 0.58±0.11 | -0.18 | 0.26 | 0.78±0.25 | 0.14 | 0.41 |
| | MAM | 0.33±0.02 | 0.42±0.13 | 0.27 | 0.43 | 0.28±0.08 | -0.21 | 0.23 | 0.35±0.10 | 0.05 | 0.26 | 0.29±0.08 | -0.13 | 0.27 | 0.31±0.05 | -0.09 | 0.18 | 0.37±0.08 | 0.10 | 0.21 |
| | SON | 0.52±0.20 | 0.49±0.12 | -0.05 | 0.18 | 0.29±0.07 | -0.78 | 0.78 | 0.39±0.13 | -0.34 | 0.34 | 0.37±0.05 | -0.26 | 0.26 | 0.46±0.11 | -0.13 | 0.13 |  |  |  |
| | DJF | 0.25±0.08 | 0.14±0.05 | -0.86 | 0.97 | 0.14±0.05 | -0.86 | 0.86 | 0.15±0.06 | -0.72 | 0.87 | 0.24±0.04 | -0.04 | 0.21 | 0.26±0.03 | 0.02 | 0.23 | 0.26±0.04 | 0.05 | 0.27 |
| ENF | JJA | 0.58±0.23 | 0.46±0.12 | -0.29 | 0.35 | 0.46±0.11 | -0.30 | 0.42 | 0.47±0.10 | -0.27 | 0.40 | 0.42±0.14 | -0.39 | 0.68 | 0.52±0.14 | -0.14 | 0.44 | 0.53±0.13 | -0.12 | 0.42 |
| | MAM | 0.46±0.15 | 0.35±0.11 | -0.31 | 0.43 | 0.34±0.10 | -0.34 | 0.40 | 0.37±0.12 | -0.24 | 0.37 | 0.42±0.06 | -0.10 | 0.31 | 0.43±0.09 | -0.07 | 0.26 | 0.46±0.11 | -0.01 | 0.26 |
| | SON | 0.47±0.22 | 0.35±0.12 | -0.35 | 0.43 | 0.28±0.07 | -0.64 | 0.68 | 0.26±0.04 | -0.83 | 0.85 | 0.39±0.13 | -0.21 | 0.46 | 0.41±0.15 | -0.13 | 0.37 | 0.40±0.12 | -0.18 | 0.43 |
| | DJF | 0.32±0.21 | 0.17±0.07 | -0.87 | 0.89 | 0.19±0.08 | -0.66 | 0.73 | 0.16±0.06 | -0.98 | 1.01 | 0.29±0.08 | -0.08 | 0.29 | 0.30±0.15 | -0.05 | 0.28 | 0.28±0.12 | -0.14 | 0.36 |
| CRO | / | 0.53±0.16 | 0.50±0.26 | -0.05 | 0.29 | 0.72±0.15 | 0.37 | 0.43 | 0.81±0.13 | 0.54 | 0.54 | 0.54±0.11 | 0.03 | 0.18 | 0.61±0.15 | 0.16 | 0.32 | 0.67±0.14 | 0.27 | 0.32 |
| TRF | / | 0.76±0.48 | 1.11±0.07 | 0.46 | 0.56 | 0.98±0.06 | 0.29 | 0.52 | 1.10±0.10 | 0.44 | 0.53 | 0.47±0.05 | -0.60 | 0.85 | 0.57±0.04 | -0.33 | 0.61 | 0.66±0.07 | -0.14 | 0.48 |
| GRA | JJA | 0.33±0.17 | 0.72±010 | 1.21 | 1.21 | 0.59±0.21 | 0.82 | 0.82 | 0.84±0.28 | 1.56 | 1.56 | 0.50±0.12 | 0.53 | 0.79 | 0.50±0.16 | 0.51 | 0.51 | 0.68±0.21 | 1.08 | 1.08 |
| | MAM | 0.39±0.13 | 0.58±0.13 | 0.48 | 0.48 | 0.43±0.00 | 0.08 | 0.28 | 0.62±0.16 | 0.57 | 0.74 | 0.42±0.11 | 0.06 | 0.48 | 0.46±0.03 | 0.17 | 0.36 | 0.62±0.15 | 0.56 | 0.72 |
| | SON | 0.30±0.06 | 0.59±0.21 | 1.00 | 1.20 | 0.46±0.22 | 0.55 | 0.78 | 0.55±0.26 | 0.88 | 1.03 | 0.42±0.29 | 0.43 | 0.76 | 0.46±0.20 | 0.54 | 0.80 | 0.54±0.22 | 0.82 | 0.95 |
| | DJF | 0.33±0.05 | 0.34±0.26 | 0.02 | 0.68 | 0.24±0.14 | -0.37 | 0.56 | 0.34±0.31 | 0.04 | 0.77 | 0.31±0.15 | -0.08 | 0.46 | 0.35±0.18 | 0.06 | 0.49 | 0.443±0.32 | 0.31 | 0.79 |

[Figure]

**Technical comments:**

Line 111: It would be helpful to include examples of the biosphere-atmosphere interactions that are lacking in CTMs

- *Thank you for the suggestion. Revised as follows:*

  *"Very few studies have addressed the atmospheric chemistry-vegetation feedbacks due to lack of representation of biosphere-atmosphere interactions in CTMs (Centoni, 2017; Lei et al., 2020). For example, $O_3$-induced vegetation damage can worsen $O_3$*

*air quality by modifying O$_3$-relevant fluxes (Monks et al., 2015; Sadiq et al., 2017; Zhou et al., 2018), and limit land carbon sink (Sitch et al., 2007; Lombardozzi et al., 2015). Two-way nitrogen exchange that includes the impacts of nitrogen deposition on soil and plant biogeochemistry and the subsequent secondary effects on atmospheric chemistry is also largely lacking (Zhao et al., 2017; Liu et al., 2021). Higher ambient CO$_2$ concentration can also affect plant stomatal conductance and photosynthesis, in turn causing changes in transpiration and hence in surface temperature, cloud cover, and meteorology (e.g., Sanderson et al., 2007)."*

Lines 140-143: There are multiple implementations of the W89 and the Z03 schemes in CTMs. Please be more specific here as to which implementation you used for both frameworks and not any specific differences that would have implications to the results here.

- *Thank you for your suggestion. Revised as suggested:*

  *"We examined two major dry deposition modeling frameworks: (1) the Wesely framework, which has been widely used in global atmospheric chemistry models (Hardacre et al., 2015; Morgenstern et al., 2017; Porter et al., 2019; Silva and Heald, 2018), and in this study we used the Wesely scheme version (referred to as W89 hereafter) as currently implemented in the GEOS-Chem chemical transport model with modification by Wang et al. (1998); (2) the Zhang et al. (2003) dry deposition framework used in several regional air quality models (Nopmongcol et al., 2012; Schwede et al., 2011; Zhang et al., 2009). Here we implemented the scheme as described in Zhang et al. (2003) (referred to as Z03 hereafter)."*

Line 170: Not specific to this line of text per se, but little is said about the stomatal blocking factor that is in the Z03 scheme and the potential effect of it. While it doesn't affect the actual value of the stomatal resistance, it definitely has implications for the contribution of that pathway to the overall canopy resistance and the deposition velocity. It merits a bit of attention in the paper.

- *Thanks for the suggestion. We conducted tests using Z03 without stomatal blocking for rainforests where the highest precipitation is expected, and Z03 simulates lower $v_d$ during the day with stomatal blocking. Stomatal blocking factor contributes little to the differences in simulated seasonal average $v_d$ rainforest between W89 and Z03. We added discussion about stomatal blocking factor in Z03:*

  *"Z03 considers stomatal blocking that occurs after rain or dew events, and thus simulates lower dry deposition velocities at measurement sites with high precipitation. However, for most observational sites used in this study, precipitation rates are lower than the stomatal blocking threshold throughout the measurement periods, and stomatal blocking contributes little to the differences in simulated $v_d$ across different schemes."*

Lines 220-226: It has previously been stated that the W89 version is the one in GEOS-Chem and it doesn't need to be restated here. The focus here is clearly on the stomatal parameterization. Have other papers already addressed the non-stomatal resistances? Perhaps cite any studies that have here and maybe make recommendations about whether that work should or could be part of future work.

- *Thank you for your suggestion. Revised as suggested:*

*"To evaluate the two dry deposition frameworks and to compare the multiplicative and photosynthesis-based stomatal schemes, we replaced the default stomatal parameterization in W89 and Z03 dry deposition frameworks with FBB and MED, and in total six dry deposition configurations were tested as described in Table 1. The differences between the W89 and Z03 frameworks lie in not only stomatal parameterization, but also non-stomatal deposition structures and algorithms. For non-stomatal resistances, Z03 considers variations from meteorological (e.g., RH, u\*) and biological factors (e.g., LAI, wet or dry canopy), while W89 uses simpler representation of cuticular resistance and aerodynamic resistance (Table 1). Mechanistic non-stomatal parameterization remains challenging due to uncertainties in inferred non-stomatal deposition estimates, such as difficulties in separating non-stomatal uptake from soil uptake and in-canopy chemistry (Clifton et al., 2020a). Future evaluation of non-stomatal algorithms requires further measurements such as BVOC emissions and soil moisture (Clifton et al., 2019)."*

Lines 227-228, it would be helpful to add "in TEMIR" after mode for clarity or you could add text to the model description to indicate that it runs both in single-site mode and as a gridded model.

- *Thank you for your suggestion. Revised as suggested.*

Lines 239-240: It would be helpful here to indicate how the model handles multiple land use types within the grid cell. Again, this could be added here or in the model description. It is stated a bit later but would be better addressed in Section 2.1. The grid cell size is pretty large for ecosystem specific studies. What impacts might that have?

- *Thank you for your suggestion. Subgrid heterogeneity such as vegetation traits is poorly represented by large grid cells in general. Simulated $v_d$ and $G_s$ values under extreme local meteorological conditions might also be flattened. These could be better resolved in future studies where models can be run efficiently at finer scales with high-resolution input data.*
- *Revised as suggested: "… For global simulations, the model was run at a spatial resolution of 2°×2.5° driven by MERRA-2 meteorology for each dry deposition configuration. $v_d$ and $G_s$ were summed up by PFT fractions over vegetated land within each grid cell."*

Line 265: It would be helpful to provide the Penman-Monteith method in the Supplemental.

- *Thanks for the suggestion. Added P-M method in the Supplementary.*

*"We use evaporative-resistance form of Penman-Monteith method to keep consistent with SynFlux stomatal conductance. The leaf stomatal conductance is:*

$$g_w^{-1} = \frac{\varepsilon\rho(e_s(T_f)-e)}{pE} - (r_a + r_{b,w}),$$

*where $\varepsilon$ is mass ratio between water and dry air, $p$ is air pressure, $E$ is surface moisture flux, $T_f$ is leaf temperature, $e_s(T_f)$ is the saturation vapor pressure at leaf surface. $r_a$ is aerodynamic resistance, $r_{b,w}$ is quasi-laminar layer resistance to water vapor. $T_f$ is estimated as follows:*

$$T_f = T + \frac{H(r_a + r_{b,H})}{c_p \rho},$$

*where T is air temperature, H is sensitive heat, $c_p$ is specific heat of air, $\rho$ is the mass density of air, $r_{b,H}$ is quasi-laminar layer resistance to heat.*

*Stomatal conductance of $O_3$ is calculated with molecular diffusion coefficient ratio 0.6 between $O_3$ and water vapor:*

$$g_s = 0.6 g_w \text{ ''}$$

Line 292-292: I don't agree with the statement that the schemes can generally capture the magnitude for the major PFTs. The models do not capture the range of values observed for the coniferous forest or the rainforest and predict a range of values for grasses that is larger than the observations.

- *Thanks for the suggestion. Revised as suggested:*

    *". The dry deposition schemes used in this study fit observed $v_d$ better for deciduous forest and crops. Yet different schemes cannot reproduce daytime $v_d$ well for coniferous forest, grass and rainforest."*

Line 305: How did you determine that the modeled vd for grasses is largely determined by the minimum stomatal resistance?

- *Thanks for your comment. Overestimation of $v_d$ compared in this study is different from previous works where grassland $v_d$ is underestimated. Revised as below:*

    *"In previous studies, models mostly underestimated grassland $v_d$ (Hardacre et al., 2015; Pio et al., 2000). Discrepancies between our modeled grassland $v_d$ and previous works mainly arise from the prescribed minimum stomatal resistance ($r_{smin}$) and LAI."*

Line 469: It might be helpful here to add a few sentences comparing these results to the ones for the field study sites.

- *Thanks for the suggestion. Revised as suggested:*

    *"...while MED overestimates $G_s$ (NMBF = 0.44), and W89 simulates with NMAEF = 0.41, lower than other schemes (NMAEF > 0.50). Evaluation with long-term measurements in Sect 3.2 finds similar model performance using different stomatal schemes. As shown in Fig. 4a and Fig. 4c, Z03 and FBB simulate comparable diurnal $G_s$ cycles. MED produces higher $G_s$ values than FBB in general."*

Line 535-536: It isn't clear to me where in Figure 9a, the non-stomatal vs stomatal deposition rates are provided.

- *Thanks for the comment. It is interpreted from Fig. 10e where Z03 has lower stomatal deposition than W89. Revised as suggested:*

    *"Z03 simulates lower daytime $v_d$ than W89 in most regions, except for evergreen needleleaf regions at high latitudes (Fig. 9a), where Z03 simulates higher stomatal*

*deposition than W89 (Fig. 10e). Hence differences in daytime $v_d$ for these regions are caused by higher non-stomatal deposition simulated by Z03."*

Line 385: Define RuBP; also is equation 8 also from Franks et al.?

- *Thanks for the suggestion. Yes, equation 8 is from Franks et al. (2018). We added definition of RuBP as suggested: RuBP (ribulose 1,5-bisphosphate)*

Figure 11: Why is W89 only used in this figure rather than including Z03?

- *Thanks for the suggestion. We assumed W89 and Z03 use the same factor (Eq. 8) to represent the relative change in stomatal conductance under rising $CO_2$ concentrations and that the simulated changes in stomatal conductance are identical. We added Z03 as suggested.*

[Figure]

Line 622-623: The results are not very convincing for switching to a photosynthesis-based model

- *Thanks for your comments. More caution is now conveyed, as stipulated above.*

Line 637: I am a curious about the use of "guaranteed" here.

- *W89 was parameterized with observations and intended to capture long-term, or seasonal, dry deposition velocities.*

Line 648: While I fundamentally agree that photosynthesis-based models offer opportunities for improving our estimates of air-surface exchange, I think there is still a lot to be considered when coupling in grid models with subgrid variability in vegetation types.

- *Thank you for your comments. More caution is now conveyed and more discussion is devoted to this, as stipulated above.*

**Editing notes:**

Lines 27 – 29: Sentence could benefit from editing

- *Revised as suggested.*

Line 89: Consider rewording to "tree and crop species of concern"

- *Revised as suggested.*

Line 90: Add "the" before DOSE

- *Revised as suggested.*

Line 127-128: Change "discussed" to "discuss"; add "the" before stomatal; parameterization should be plural

- *Revised as suggested.*

Line 157: add "the" before dry

- *Revised as suggested.*

Line 179: change its to their

- *Revised as suggested.*

Line 194: change "is" to are

- *Revised as suggested.*

Equation 4: the concentration should be denoted by C not c

- *Revised as suggested.*

Line 213: add the before photosynthetic

- *Revised as suggested.*

Line 277: Insert "the" before different and dry

- *Revised as suggested.*

Figure 1: in the caption, add "where" before different. I would also suggest using different colors as the current ones may not be optimal for users with color blindness issues.

- *Thanks for the suggestion. Revised as suggested. Figure plotted with "Okabe-Ito" palette (https://jfly.uni-koeln.de/color/) which is friendly to colorblind people.*

Line 352: add "mean: after monthly

- *Revised as suggested.*

Line 360: delete "in this"

- *Revised as suggested.*

Line 377: Perhaps the second vd in the sentence should be Gs?

- *Yes. Revised as suggested to be precise.*

Line 604: I have seen those experiments referred to as FACE not FREE

- *Thank you. Corrected as suggested.*

**References**

Clifton, O. E., Fiore, A. M., Munger, J. W., & Wehr, R.: Spatiotemporal controls on observed daytime ozone deposition velocity over Northeastern U.S. forests during summer, J. Geophys. Res.-Atmos, 124, 5612–5628, https://doi.org/10.1029/2018JD029073, 2019.

Liu, X., Tai, A.P.K., Chen, Y. et al. Publisher Correction: Dietary shifts can reduce premature deaths related to particulate matter pollution in China, Nat. Food, https://doi.org/10.1038/s43016-022-00458-2, 2022.

Lombardozzi, D., Levis, S., Bonan, G., Hess, P. G., & Sparks, J. P.: The influence of chronic ozone exposure on global carbon and water cycle, J. Climate, 28(1), 292–305, https://doi.org/10.1175/Jcli-D-14-00223.1, 2015.

Monks, P. S., Archibald, A. T., Colette, A., Cooper, O., Coyle, M., Derwent, R., Fowler, D., Granier, C., Law, K. S., Mills, G. E., Stevenson, D. S., Tarasova, O., Thouret, V., von Schneidemesser, E., Sommariva, R., Wild, O., and Williams, M. L.: Tropospheric ozone and its precursors from the urban to the global scale from air quality to short-lived climate forcer, Atmos. Chem. Phys., 15, 8889–8973, https://doi.org/10.5194/acp-15-8889-2015, 2015.

Sadiq, M., Tai, A. P. K., Lombardozzi, D., & Martin, M. V.: Effects of ozone-vegetation coupling on surface ozone air quality via biogeochemical and meteorological feedbacks. Atmos. Chem. Phys., 17(4), 3055-3066. https://doi.org/10.5194/acp-17-3055-2017, 2017.

Sitch, S., Cox, P. M., Collins, W. J., & Huntingford, C.: Indirect radiative forcing of climate change through ozone effects on the land-carbon sink, Nature, 448(7155), 791-U794, http://doi.org/10.1038/nature06059, 2007.

Zhao, Y., Zhang, L., Tai, A. P. K., Chen, Y., and Pan, Y.: Responses of surface ozone air quality to anthropogenic nitrogen deposition in the Northern Hemisphere, Atmos. Chem. Phys., 17, 9781–9796, https://doi.org/10.5194/acp-17-9781-2017, 2017.

Zhou, S. S., Tai, A. P. K., Sun, S. H., Sadiq, M., Heald, C. L., & Geddes, J. A.: Coupling between surface ozone and leaf area index in a chemical transport model: strength of feedback and implications for ozone air quality and vegetation health, Atmos. Chem. Phys., 18(19), 14133-14148. doi:10.5194/acp-18-14133-2018, 2018.

---

## Author Response (AR2)

**Response to Reviewers' Comments on "Influence of plant ecophysiology on ozone dry deposition: Comparing between multiplicative and photosynthesis-based dry deposition schemes and their responses to rising CO₂ level" by Sun et al.**

The comments of the referee are given as plain text, while the authors' response is given in *italic*. We have revised the paper based on the reviewers' comments.

***Response to Referee #1***

In this paper, the authors use a standalone terrestrial biosphere model to evaluate both multiplicative and photosynthesis-based schemes of stomatal conductance of ozone. Observational datasets of the dry deposition velocity and the stomatal conductance of ozone are used to do the model evaluation. The authors suggested that the photosynthesis-based stomatal algorithms that captured the responses to water stress had a better agreement with the observations. The manuscript describes a straightforward modeling study exploring basic parameterizations and comparisons to observations, and fits into the scope of Biogeosciences. I have a few minor comments as listed below.

- *We thank the referee for the very helpful comments and suggestions. The paper has been revised accordingly. Our point-by-point responses are provided below.*

My major concern is that based on the model-observation comparison in this paper, I do not see a significant improvement by using photosynthesis-based stomatal conductance methods, compared to the traditional multiplicative methods. The default multiplicative W89 scheme without stomatal response to water stress fails to reproduce the diurnal variations in Gs, but the multiplicative Z03 method seems to agree well with the other photosynthesis-based methods and the observations. Furthermore, all schemes compare poorly with observations in rainforests and in the Blodgett forest site (which is often associated with higher temperatures and water stress). Can the authors comment a bit more on the advantages of using photosynthesis-based methods?

- *We very much agree that using photosynthesis-based stomatal conductance models do not significantly improve model performance over Z03, and that Z03 agrees well with photosynthesis-based methods. In the paper, we have therefore mentioned this at various places, highlighting that both photosynthesis-based schemes and Z03 multiplicative schemes are better than W89, mostly likely due to their ability to capture plant responses to water stress and VPD, e.g., P17 L501: "... In general, accounting for stomatal response to VPD and/or water stress using multiplicative or photosynthesis-based stomatal algorithms can improve model performance in capturing diurnal variations of $G_s$ and $v_d$."*
- *The merits of multiplicative methods such as lower computational costs and higher compatibility are undeniable for Earth system modeling. Multiplicative methods parameterized with observations can also be improved whenever more field measurements are available. Yet, with more biophysically meaningful and measurable properties, photosynthesis-based methods are principally more mechanistic than multiplicative stomatal methods, and can better address plant responses to a changing environment (e.g., rising CO₂ and temperature) with rapidly expanding*

*knowledge from biologists. Furthermore, parameters for photosynthesis-based methods can be obtained from leaf-scale measurements, which overall cost less than dry deposition flux measurements that are used for parameterizing multiplicative model. We have now emphasized these points more in the Conclusions and Discussion, e.g., P28 L750: "Our attempt to include the empirical $CO_2$ response function of Franks et al. (2013) in multiplicative stomatal schemes result in a much larger reduction in global $G_s$ that doubled the average relative change computed with photosynthesis-based stomatal schemes, and potentially overstates stomatal responses to elevated $CO_2$ under future scenarios."*

The numbers and names of the modeling schemes are sometimes confusing. For example, the words "multiplicative" and "photosynthesis-based" in the title refer to stomatal conductance schemes, not dry deposition schemes, right? In the abstract, the Medlyn scheme is also a photosynthesis-based method, so there are actually two multiplicative (W89, Z03), two photosynthesis-based (FBB, MED) stomatal conductance schemes. I think it should be stated clearly in the abstract and introduction, or it will confuse the readers.

- *Thanks for your suggestions. Yes, "multiplicative" and "photosynthesis-based" refer to stomatal conductance schemes. The relevant parts are revised accordingly.*

  *P1 L16: "We developed and used a standalone terrestrial biosphere model, driven by a unified set of prescribed meteorology, to evaluate two widely used dry deposition modeling frameworks, Wesely (1989) and Zhang et al. (2003), with different configurations of stomatal resistance: 1) the default multiplicative method in the Wesely scheme (W89) and Zhang et al. (2003) scheme (Z03); 2) the traditional photosynthesis-based Farquhar-Ball-Berry (FBB) stomatal algorithm; 3) the Medlyn stomatal algorithm (MED) based on optimization theory."*

Also, the figures should be consistent to show all 6 schemes when comparing to observed dry deposition velocity Vd, and show all 4 schemes when comparing to observed stomatal conductance Gs. For example, why not compare the Z03 scheme in Figure 11?

- *Thank you for your suggestion. We updated the Figure 11 to make it consistent.*

L42: Does this "45%" refer to an annually averaged percentage? How does this compare to your results? As the stomatal conductance is the main focus of this paper, I would suggest moving Figure S3 (showing the fraction of stomatal conductance to total deposition) to the main text.

- *Thanks for the suggestion. Yes "45%" refers to annual daytime average (Clifton et al., 2020), which is stomatal fraction of ozone dry deposition aggregated from previous literature. We aggregate annual daytime stomatal fraction here using SynFlux: W89 (87%), Z03 (62%), FBB (65%), MED (68%). Stomatal fractions in Clifton et al. (2020) are calculated with P-M method. Our results show that Z03 and FBB agree with P-M derived stomatal conductance. Simulated higher stomatal fractions can be related with underestimation of non-stomatal conductance. Not all datasets in Clifton et al. include data from all seasons. The magnitude of stomatal fraction is also*

*affected by vegetation types: deciduous forest has higher stomatal fraction than other vegetation types. We have now revised the relevant parts and moved Figure S3 to main text as suggested and added corresponding description of Figure S3 in main text: P14 L440: "Figure 5 shows the fractions of monthly average daytime stomatal conductance to canopy conductance ($G_c = 1/R_c$), and that higher fractions indicate higher ratios of stomatal deposition to non-stomatal deposition."*

L257: Please briefly explain the P-M method here.

- *We have now explained the P-M method in the Supplementary Text S3:*

   *"We use evaporative-resistance form of Penman-Monteith method to keep consistent with SynFlux stomatal conductance. The leaf stomatal conductance is:*

   $$g_w^{-1} = \frac{\varepsilon\rho(e_s(T_f)-e)}{pE} - (r_a + r_{b,w}),$$

   *where $\varepsilon$ is mass ratio between water and dry air, p is air pressure, E is surface moisture flux, $T_f$ is leaf temperature, $e_s(T_f)$ is the saturation vapor pressure at leaf surface. $r_a$ is aerodynamic resistance, $r_{b,w}$ is quasi-laminar layer resistance to water vapor. $T_f$ is estimated as follows:*

   $$T_f = T + \frac{H(r_a+r_{b,H})}{c_p\rho},$$

   *where T is air temperature, H is sensitive heat, $c_p$ is specific heat of air, $\rho$ is the mass density of air, $r_{b,H}$ is quasi-laminar layer resistance to heat.*

   *Stomatal conductance of $O_3$ is calculated with molecular diffusion coefficient ratio 0.6 between $O_3$ and water vapor:*

   $$g_s = 0.6g_w \quad "$$

Table 3: This table contains a lot information and is not easy to read. How about using some background colors, e.g., red/blue to show overestimation/underestimation and dark/light colors to indicate large/small bias?

- *Thank you for the suggestion. Updated Table 3 with background colors.*

L327 Not sure what this sentence means. Do you mean ozone reacts "with" soil-emitted NO and BVOC here?

- *Yes. Revised as suggested:*

   *P12 L382 "Non-stomatal $O_3$ deposition includes chemical reactions of $O_3$ with nitric oxide (NO) and biogenic volatile organic compounds (BVOC) from soil emissions (Fares et al., 2012)."*

Figure 2 The models seem to predict an overall earlier peak than the observations. Can the authors comment on why it could be?

- *Simulated monthly daytime $v_d$ peaks in around June to July, while observed daytime $v_d$ peaks during July to August. As we used observed LAI in this study, LAI is not the major driver as in previous studies. For long-term sites in Figure 2, overestimation of stomatal conductance and underestimation of non-stomatal conductance cause discrepancies between modelled and observed $v_d$. Simulated and observed monthly daytime average stomatal conductance variations are shown in the figure below. The peaks in early summer are mainly driven by stomatal conductance, due to favorable conditions such as higher solar radiation and lower VPD.*

[Figure]

L401 which site is "ponderosa pine forest"? Include the site name here.

- *Blodgett Ameriflux site. Revised as suggested:*

    *P16 L468: "The major $O_3$ removal process in the ponderosa pine plantation at the Blodegett Ameriflux site is non-stomatal $O_3$ sink through in-canopy chemical reactions between $O_3$ and BVOC (Fares et al., 2010; Kurpius and Goldstein, 2003)."*

Finally, this manuscript includes many abbreviations and sometimes is hard to follow. I would suggest including a list of abbreviations and explanations if possible.
- *Thanks for your suggestion. Revised as suggested. We added a table of abbreviations and explanations in the supplementary.*

The authors examine the impact of carrying the stomatal conductance parameterization on the predicted deposition velocity. Comparisons are made between the well-used Wesely (1989) and Zhang et al. (2003) approaches as well as derivative parameterizations developed by substituting two different photosynthesis-based approaches into both the Wesely and Zhang approaches.   Results are compared against observational data from several flux studies as well as the SynFlux dataset.   Overall, the paper is sound but could benefit from additional editing.   The text in Table 3 is very small as is the text in Figure 1.   I am surprised that the photosynthetic models did not offer greater improvements in modeled values and would be interested in your thoughts on that.   I would also be interested in seeing values of the correlation coefficient.

- *We thank the reviewer for the insightful comments and suggestions. The manuscript has been revised accordingly.*

- *We agree that photosynthesis-based stomatal conductance models do not significantly improve model performance. In the paper, we have therefore mentioned this at various places, highlighting that both photosynthesis-based schemes and Z03 multiplicative schemes are better than W89, mostly likely due to their ability to capture plant responses to water stress and VPD, e.g., P17 L501: "... In general, accounting for stomatal response to VPD and/or water stress using multiplicative or photosynthesis-based stomatal algorithms can improve model performance in capturing diurnal variations of $G_s$ and $v_d$."*

- *The merits of multiplicative methods such as lower computational costs and higher compatibility are undeniable for Earth system modeling. Multiplicative methods parameterized with observations can also be improved whenever more field measurements are available. Yet, with more biophysically meaningful and measurable properties, photosynthesis-based methods are principally more mechanistic than multiplicative stomatal methods, and can better address plant responses to a changing environment (e.g., rising $CO_2$ and temperature) with rapidly expanding knowledge from biologists. Furthermore, parameters for photosynthesis-based methods can be obtained from leaf-scale measurements, which overall cost less than dry deposition flux measurements that are used for parameterizing multiplicative model. We have now emphasized these points more in the Conclusions and Discussion, e.g., P28 L750: "Our attempt to include the empirical $CO_2$ response function of Franks et al. (2013) in multiplicative stomatal schemes result in a much larger reduction in global $G_s$ that doubled the average relative change computed with photosynthesis-based stomatal schemes, and potentially overstates stomatal responses to elevated $CO_2$ under future scenarios."*

*We calculated Pearson correlation coefficient R for seasonal average modelled and observed $v_d$ at all observational sites: W89 (R = 0.61), W89FBB (R = 0.56), W89MED (R = 0.48), Z03 (R = 0.37), Z03FBB (R = 0.47), Z03MED (R = 0.41). The p-values for all coefficients here are <0.01.*

*We updated font size of Table 3 and Figure 1 shown as below:*

| PFT | Season | Observation | W89 | | | W89FBB | | | W89MED | | | Z03 | | | Z03FBB | | | Z03MED | | |
|---|---|---|---|---|---|---|---|---|---|---|---|---|---|---|---|---|---|---|---|---|
| | | mean±sd | mean±sd | NMBF | NMAEF | mean±sd | NMBF | NMAEF | mean±sd | NMBF | NMAEF | mean±sd | NMBF | NMAEF | mean±sd | NMBF | NMAEF | mean±sd | NMBF | NMAEF |
| DBF | JJA | 0.69±0.10 | 0.90±0.17 | 0.32 | 0.32 | 0.59±0.10 | −0.16 | 0.26 | 0.81±0.24 | 0.18 | 0.41 | 0.55±0.09 | −0.25 | 0.30 | 0.58±0.11 | −0.18 | 0.26 | 0.78±0.25 | 0.14 | 0.41 |
| | MAM | 0.33±0.02 | 0.42±0.13 | 0.27 | 0.43 | 0.28±0.08 | −0.21 | 0.23 | 0.35±0.10 | 0.05 | 0.26 | 0.29±0.08 | −0.13 | 0.27 | 0.31±0.05 | −0.09 | 0.18 | 0.37±0.08 | 0.10 | 0.21 |
| | SON | 0.52±0.20 | 0.49±0.12 | −0.05 | 0.18 | 0.29±0.07 | −0.78 | 0.78 | 0.39±0.13 | −0.34 | 0.34 | 0.41±0.06 | −0.26 | 0.26 | 0.37±0.05 | −0.39 | 0.39 | 0.46±0.11 | −0.11 | 0.13 |
| | DJF | 0.25±0.08 | 0.14±0.05 | −0.86 | 0.97 | 0.14±0.05 | −0.86 | 0.86 | 0.15±0.06 | −0.72 | 0.87 | 0.24±0.04 | −0.04 | 0.21 | 0.26±0.03 | 0.02 | 0.23 | 0.26±0.04 | 0.05 | 0.27 |
| ENF | JJA | 0.58±0.23 | 0.46±0.12 | −0.29 | 0.35 | 0.46±0.11 | −0.30 | 0.42 | 0.47±0.10 | −0.27 | 0.40 | 0.42±0.14 | −0.39 | 0.68 | 0.52±0.14 | −0.14 | 0.44 | 0.53±0.13 | −0.12 | 0.42 |
| | MAM | 0.46±0.15 | 0.35±0.11 | −0.31 | 0.43 | 0.34±0.10 | −0.34 | 0.40 | 0.37±0.10 | −0.24 | 0.37 | 0.42±0.06 | −0.10 | 0.31 | 0.43±0.09 | −0.07 | 0.26 | 0.46±0.11 | −0.01 | 0.26 |
| | SON | 0.47±0.22 | 0.35±0.12 | −0.35 | 0.43 | 0.28±0.07 | −0.64 | 0.68 | 0.26±0.04 | −0.83 | 0.85 | 0.39±0.13 | −0.21 | 0.46 | 0.41±0.15 | −0.13 | 0.37 | 0.40±0.12 | −0.18 | 0.43 |
| | DJF | 0.32±0.21 | 0.17±0.07 | −0.87 | 0.89 | 0.19±0.08 | −0.66 | 0.73 | 0.16±0.06 | −0.98 | 1.01 | 0.30±0.11 | −0.08 | 0.29 | 0.30±0.15 | −0.05 | 0.28 | 0.28±0.12 | −0.14 | 0.36 |
| CRO | / | 0.53±0.16 | 0.50±0.26 | −0.05 | 0.29 | 0.72±0.15 | 0.37 | 0.43 | 0.81±0.13 | 0.54 | 0.54 | 0.54±0.11 | 0.03 | 0.18 | 0.61±0.15 | 0.16 | 0.32 | 0.67±0.14 | 0.27 | 0.32 |
| TRF | / | 0.76±0.48 | 1.11±0.07 | 0.46 | 0.56 | 0.98±0.06 | 0.29 | 0.52 | 1.10±0.10 | 0.44 | 0.53 | 0.47±0.05 | −0.60 | 0.85 | 0.57±0.04 | −0.33 | 0.61 | 0.66±0.07 | −0.14 | 0.48 |
| GRA | JJA | 0.33±0.17 | 0.72±010 | 1.21 | 1.21 | 0.59±0.21 | 0.82 | 0.82 | 0.84±0.28 | 1.56 | 1.56 | 0.50±0.12 | 0.53 | 0.79 | 0.50±0.16 | 0.51 | 0.51 | 0.68±0.21 | 1.08 | 1.08 |
| | MAM | 0.39±0.13 | 0.58±0.13 | 0.48 | 0.48 | 0.43±0.00 | 0.08 | 0.28 | 0.62±0.16 | 0.57 | 0.74 | 0.42±0.11 | 0.06 | 0.48 | 0.46±0.03 | 0.17 | 0.36 | 0.62±0.15 | 0.56 | 0.72 |
| | SON | 0.30±0.06 | 0.59±0.21 | 1.00 | 1.20 | 0.46±0.22 | 0.55 | 0.78 | 0.55±0.26 | 0.88 | 1.03 | 0.42±0.29 | 0.43 | 0.76 | 0.46±0.20 | 0.54 | 0.80 | 0.54±0.22 | 0.82 | 0.95 |
| | DJF | 0.33±0.05 | 0.34±0.26 | 0.02 | 0.68 | 0.24±0.14 | −0.37 | 0.56 | 0.34±0.31 | 0.04 | 0.77 | 0.31±0.15 | −0.08 | 0.46 | 0.35±0.18 | 0.06 | 0.49 | 0.443±0.32 | 0.31 | 0.79 |

[Figure]

**Technical comments:**

Line 111: It would be helpful to include examples of the biosphere-atmosphere interactions that are lacking in CTMs

- *Thank you for the suggestion. Revised as follows:*

  *P3 L120: "Very few studies have addressed the atmospheric chemistry-vegetation feedbacks due to lack of representation of biosphere-atmosphere interactions in CTMs (Centoni, 2017; Lei et al., 2020). For example, $O_3$-induced vegetation damage can worsen $O_3$ air quality by modifying $O_3$-relevant fluxes (Monks et al., 2015; Sadiq et al., 2017; Zhou et al., 2018; Zhu et al., 2022), and limit land carbon sink (Sitch et al., 2007; Lombardozzi et al., 2015). Two-way nitrogen exchange that includes the impacts of nitrogen deposition on soil and plant biogeochemistry and the subsequent secondary effects on atmospheric chemistry is also largely lacking (Zhao et al., 2017; Liu et al., 2022). Higher ambient $CO_2$ concentration can also affect plant stomatal conductance and photosynthesis, in turn causing changes in transpiration and hence in surface temperature, cloud cover, and meteorology (e.g., Sanderson et al., 2007)."*

Lines 140-143: There are multiple implementations of the W89 and the Z03 schemes in CTMs. Please be more specific here as to which implementation you used for both frameworks and not any specific differences that would have implications to the results here.

- *Thank you for your suggestion. Revised as suggested:*

  *P4 L155: "We examined two major dry deposition modeling frameworks: (1) the Wesely framework, which has been widely used in global atmospheric chemistry models (Hardacre et al., 2015; Morgenstern et al., 2017; Porter et al., 2019; Silva and Heald, 2018), and in this study we used the Wesely scheme version (referred to as W89 hereafter) as currently implemented in the GEOS-Chem chemical transport model with modification by Wang et al. (1998); (2) the Zhang et al. (2003) dry deposition framework used in several regional air quality models (Nopmongcol et al., 2012; Schwede et al., 2011; Zhang et al., 2009). Here we implemented the scheme as described in Zhang et al. (2003) (referred to as Z03 hereafter)."*

Line 170: Not specific to this line of text per se, but little is said about the stomatal blocking factor that is in the Z03 scheme and the potential effect of it. While it doesn't affect the actual value of the stomatal resistance, it definitely has implications for the contribution of that pathway to the overall canopy resistance and the deposition velocity. It merits a bit of attention in the paper.

- *Thanks for the suggestion. We conducted tests using Z03 without stomatal blocking for rainforests where the highest precipitation is expected, and Z03 simulates lower $v_d$ during the day with stomatal blocking. Stomatal blocking factor contributes little to the differences in simulated seasonal average $v_d$ rainforest between W89 and Z03. We added discussion about stomatal blocking factor in Z03:*

  *P17 L511: "Z03 considers stomatal blocking that occurs after rain or dew events, and thus simulates lower dry deposition velocities at measurement sites with high precipitation. However, for most observational sites used in this study, precipitation rates are lower than the stomatal blocking*

*threshold throughout the measurement periods, and stomatal blocking contributes little to the differences in simulated $v_d$ across different schemes."*

Lines 220-226: It has previously been stated that the W89 version is the one in GEOS-Chem and it doesn't need to be restated here. The focus here is clearly on the stomatal parameterization. Have other papers already addressed the non-stomatal resistances? Perhaps cite any studies that have here and maybe make recommendations about whether that work should or could be part of future work.

- *Thank you for your suggestion. Revised as suggested:*

  *P7 L253: "To evaluate the two dry deposition frameworks and to compare the multiplicative and photosynthesis-based stomatal schemes, we replaced the default stomatal parameterization in W89 and Z03 dry deposition frameworks with FBB and MED, and in total six dry deposition configurations were tested as described in Table 1. The differences between the W89 and Z03 frameworks lie in not only stomatal parameterization, but also non-stomatal deposition structures and algorithms. For non-stomatal resistances, Z03 considers variations from meteorological (e.g., RH, $u_*$) and biological factors (e.g., LAI, wet or dry canopy), while W89 uses simpler representation of cuticular resistance and aerodynamic resistance (Table 1). Mechanistic non-stomatal parameterization remains challenging due to uncertainties in inferred non-stomatal deposition estimates, such as difficulties in separating non-stomatal uptake from soil uptake and in-canopy chemistry (Clifton et al., 2020a). Future evaluation of non-stomatal algorithms requires further measurements such as BVOC emissions and soil moisture (Clifton et al., 2019)."*

Lines 227-228, it would be helpful to add "in TEMIR" after mode for clarity or you could add text to the model description to indicate that it runs both in single-site mode and as a gridded model.

- *Thank you for your suggestion. Revised as suggested. P7 L265: "Simulations using each dry deposition configuration were conducted in the single-site mode in TEMIR for the observational sites listed in Supplementary Table S1."*

Lines 239-240: It would be helpful here to indicate how the model handles multiple land use types within the grid cell. Again, this could be added here or in the model description. It is stated a bit later but would be better addressed in Section 2.1. The grid cell size is pretty large for ecosystem specific studies. What impacts might that have?

- *Thank you for your suggestion. Subgrid heterogeneity such as vegetation traits is poorly represented by large grid cells in general. Simulated $v_d$ and $G_s$ values under extreme local meteorological conditions might also be flattened. These could be better resolved in future studies where models can be run efficiently at finer scales with high-resolution input data.*

  *Revised as suggested: P7 L279: "… For global simulations, the model was run at a spatial resolution of 2°×2.5° driven by MERRA-2 meteorology for each dry deposition configuration. $v_d$ and $G_s$ were summed up by PFT fractions over vegetated land within each grid cell."*

Line 265: It would be helpful to provide the Penman-Monteith method in the Supplemental.

- *Thanks for the suggestion. We added P-M method in the Supplementary Text S3:*

    *"We use evaporative-resistance form of Penman-Monteith method to keep consistent with SynFlux stomatal conductance. The leaf stomatal conductance is:*

    $$g_w^{-1} = \frac{\varepsilon \rho (e_s(T_f) - e)}{pE} - (r_a + r_{b,w}),$$

    *where $\varepsilon$ is mass ratio between water and dry air, $p$ is air pressure, $E$ is surface moisture flux, $T_f$ is leaf temperature, $e_s(T_f)$ is the saturation vapor pressure at leaf surface. $r_a$ is aerodynamic resistance, $r_{b,w}$ is quasi-laminar layer resistance to water vapor. $T_f$ is estimated as follows:*

    $$T_f = T + \frac{H(r_a + r_{b,H})}{c_p \rho},$$

    *where $T$ is air temperature, $H$ is sensitive heat, $c_p$ is specific heat of air, $\rho$ is the mass density of air, $r_{b,H}$ is quasi-laminar layer resistance to heat.*

    *Stomatal conductance of $O_3$ is calculated with molecular diffusion coefficient ratio 0.6 between $O_3$ and water vapor:*

    $$g_s = 0.6 g_w \quad "$$

Line 292-292:  I don't agree with the statement that the schemes can generally capture the magnitude for the major PFTs.  The models do not capture the range of values observed for the coniferous forest or the rainforest and predict a range of values for grasses that is larger than the observations.

- *Thanks for the suggestion. Revised as suggested:*

    *P9 L338: ". The dry deposition schemes used in this study fit observed $v_d$ better for deciduous forest and crops. Yet different schemes cannot reproduce daytime $v_d$ well for coniferous forest, grass and rainforest."*

Line 305:  How did you determine that the modeled vd for grasses is largely determined by the minimum stomatal resistance?

- *Thanks for your comment. Overestimation of $v_d$ compared in this study is different from previous works where grassland $v_d$ is underestimated. Revised as below:*

*P9 L350: "In previous studies, models mostly underestimated grassland $v_d$ (Hardacre et al., 2015; Pio et al., 2000). Discrepancies between our modeled grassland $v_d$ and previous works mainly arise from the prescribed minimum stomatal resistance ($r_{smin}$) and LAI."*

Line 469:   It might be helpful here to add a few sentences comparing these results to the ones for the field study sites.

- *Thanks for the suggestion. Revised as suggested:*

  *P18 L545: "...while MED overestimates $G_s$ (NMBF = 0.44), and W89 simulates with NMAEF = 0.41, lower than other schemes (NMAEF > 0.50). Evaluation with long-term measurements in Sect 3.2 finds similar model performance using different stomatal schemes. As shown in Fig. 4a and Fig. 4c, Z03 and FBB simulate comparable diurnal $G_s$ cycles. MED produces higher $G_s$ values than FBB in general."*

Line 535-536: It isn't clear to me where in Figure 9a, the non-stomatal vs stomatal deposition rates are provided.

- *Thanks for the comment. It is interpreted from Fig. 10e where Z03 has lower stomatal deposition than W89. Revised as suggested:*

  *P23 L631: "Z03 simulates lower daytime $v_d$ than W89 in most regions, except for evergreen needleleaf regions at high latitudes (Fig. 9a), where Z03 simulates higher stomatal deposition than W89 (Fig. 10e). Hence differences in daytime $v_d$ for these regions are caused by higher non-stomatal deposition simulated by Z03."*

Line 385: Define RuBP; also is equation 8 also from Franks et al.?

- *Thanks for the suggestion. Yes, equation 8 is from Franks et al. (2018). We added definition of RuBP as suggested:*

  *P26 L695: "... RuBP (ribulose 1,5-bisphosphate)..."*

Figure 11: Why is W89 only used in this figure rather than including Z03?

- *Thanks for the suggestion. We assumed W89 and Z03 use the same factor (Eq. 8) to represent the relative change in stomatal conductance under rising $CO_2$ concentrations and that the simulated changes in stomatal conductance are identical. We added Z03 as suggested.*

Line 622-623: The results are not very convincing for switching to a photosynthesis-based model

- *Thanks for your comments. More caution is now conveyed, as stipulated above.*

Line 637: I am a curious about the use of "guaranteed" here.

- *W89 was parameterized with observations and intended to capture long-term, or seasonal, dry deposition velocities.*

Line 648: While I fundamentally agree that photosynthesis-based models offer opportunities for improving our estimates of air-surface exchange, I think there is still a lot to be considered when coupling in grid models with subgrid variability in vegetation types.

- *Thank you for your comments. More caution is now conveyed and more discussion is devoted to this, as stipulated above.*

**Editing notes:**

Lines 27 – 29: Sentence could benefit from editing

- *Revised as suggested:*

  *P1 L27: "Large discrepancies were also found in stomatal responses to rising $CO_2$ levels from 390 ppm to 550 ppm: multiplicative stomatal method with an empirical $CO_2$ response function produces reduction (–35%) in global stomatal conductance on average, much larger than that with photosynthesis-based stomatal method (–14–19%)."*

Line 89: Consider rewording to "tree and crop species of concern"

- *Revised as suggested. P3 L98: "… predict $O_3$ damage for concerned tree and crop species of concern…"*

Line 90: Add "the" before DOSE

- *Revised as suggested. P3 L99: "… the $DO_3SE$ model…"*

Line 127-128: Change "discussed" to "discuss"; add "the" before stomatal; parameterization should be plural

- *Revised as suggested. P4 L143: "We further discuss the importance of the stomatal algorithm in dry deposition parameterizations under elevated ambient $CO_2$ levels in atmospheric chemistry or air quality models."*

Line 157: add "the" before dry

- *Revised as suggested. P5 L184: "… on the dry deposition velocity of $O_3$…"*

Line 179: change its to their

- *Revised as suggested. P6 L206: "… whereby plants optimize  their stomatal behavior…"*

Line 194: change "is" to are

- *Revised as suggested. P6 L221: "Details of Eq. (2)  are described in Supplementary Text S2."*

Equation 4: the concentration should be denoted by C not c

- *Revised as suggested. P6 L233:* $g_s = \dfrac{1}{r_s} = g_{1B}\dfrac{A_n h_s}{C_s} + g_0,$

Line 213: add the before photosynthetic

- *Revised as suggested. P7 L246: "… leaf stomatal conductance is coupled to the photosynthetic rate…"*

Line 277: Insert "the" before different and dry

- *Revised as suggested. P9 L323: "… The simulated seasonal average daytime $v_d$ using the different dry deposition schemes…"*

Figure 1: in the caption, add "where" before different.   I would also suggest using different colors as the current ones may not be optimal for users with color blindness issues.

- *Thanks for the suggestion. Revised as suggested. P11 L370: "… except that for crops where different colors indicate crop types". All figures now are plotted with "Okabe-Ito" palette ([https://jfly.uni-koeln.de/color/)](https://jfly.uni-koeln.de/color/)) which is friendly to colorblind people.*

Line 352: add "mean: after monthly

- *Revised as suggested. P12 L407: "Figure 2 shows observed and simulated monthly mean daytime (6:00am~18:00pm) $v_d$ …"*

Line 360: delete "in this"

- *Revised as suggested. P13 L417: "different dry deposition schemes in the following  section"*

Line 377: Perhaps the second vd in the sentence should be Gs?

- *Yes. Revised as suggested to be precise. P14 L436: "… which is mainly caused by overestimated afternoon  $G_s$."*

Line 604:   I have seen those experiments referred to as FACE not FREE

- *Thank you. Corrected as suggested. P26 L715: "… Free-Air $CO_2$ Enrichment (FACE) experiments…"*

*List of all relative changes in the manuscript, page and line numbers are in accordance with the marked-up version of manuscript:*

**P1 L16:** *"We developed and used a standalone terrestrial biosphere model, driven by a unified set of prescribed meteorology, to evaluate two widely used dry deposition modeling frameworks, Wesely (1989) and Zhang et al. (2003), with different configurations of stomatal resistance: 1) the default multiplicative method in the Wesely scheme (W89) and Zhang et al. (2003) scheme (Z03); 2) the traditional photosynthesis-based Farquhar-Ball-Berry (FBB) stomatal algorithm; 3) the Medlyn stomatal algorithm (MED) based on optimization theory."*

**P1 L27:** *"Large discrepancies were also found in stomatal responses to rising $CO_2$ levels from 390 ppm to 550 ppm: multiplicative stomatal method with an empirical $CO_2$ response function produces reduction (−35%) in global stomatal conductance on average, much larger than that with photosynthesis-based stomatal method (−14–19%)."*

**P3 L98:** *"... predict $O_3$ damage for concerned tree and crop species of concern..."*

**P3 L99:** *"... the $DO_3SE$ model..."*

**P3 L120:** *"Very few studies have addressed the atmospheric chemistry-vegetation feedbacks due to lack of representation of biosphere-atmosphere interactions in CTMs (Centoni, 2017; Lei et al., 2020). For example, $O_3$-induced vegetation damage can worsen $O_3$ air quality by modifying $O_3$-relevant fluxes (Monks et al., 2015; Sadiq et al., 2017; Zhou et al., 2018; Zhu et al., 2022), and limit land carbon sink (Sitch et al., 2007; Lombardozzi et al., 2015). Two-way nitrogen exchange that includes the impacts of nitrogen deposition on soil and plant biogeochemistry and the subsequent secondary effects on atmospheric chemistry is also largely lacking (Zhao et al., 2017; Liu et al., 2022). Higher ambient $CO_2$ concentration can also affect plant stomatal conductance and photosynthesis, in turn causing changes in transpiration and hence in surface temperature, cloud cover, and meteorology (e.g., Sanderson et al., 2007)."*

**P4 L143:** *"We further discuss the importance of the stomatal algorithm in dry deposition parameterizations under elevated ambient $CO_2$ levels in atmospheric chemistry or air quality models."*

**P4 L155:** *"We examined two major dry deposition modeling frameworks: (1) the Wesely framework, which has been widely used in global atmospheric chemistry models (Hardacre et al., 2015; Morgenstern et al., 2017; Porter et al., 2019; Silva and Heald, 2018), and in this study we used the Wesely scheme version (referred to as W89 hereafter) as currently implemented in the GEOS-Chem chemical transport model with modification by Wang et al. (1998); (2) the Zhang et al. (2003) dry deposition framework used in several regional air quality models (Nopmongcol et al., 2012; Schwede et al., 2011; Zhang et al., 2009). Here we implemented the scheme as described in Zhang et al. (2003) (referred to as Z03 hereafter)."*

**P5 L184:** *"... on the dry deposition velocity of $O_3$..."*

***P6 L206:*** *"... whereby plants optimize  their stomatal behavior..."*

***P6 L221:*** *"Details of Eq. (2)  are described in Supplementary Text S2."*

***P6 L233:*** $g_s = \dfrac{1}{r_s} = g_{1B}\dfrac{A_n h_s}{C_s} + g_0,$

***P7 L246:*** *"... leaf stomatal conductance is coupled to the photosynthetic rate..."*

***P7 L253:*** *"To evaluate the two dry deposition frameworks and to compare the multiplicative and photosynthesis-based stomatal schemes, we replaced the default stomatal parameterization in W89 and Z03 dry deposition frameworks with FBB and MED, and in total six dry deposition configurations were tested as described in Table 1. The differences between the W89 and Z03 frameworks lie in not only stomatal parameterization, but also non-stomatal deposition structures and algorithms. For non-stomatal resistances, Z03 considers variations from meteorological (e.g., RH, $u_*$) and biological factors (e.g., LAI, wet or dry canopy), while W89 uses simpler representation of cuticular resistance and aerodynamic resistance (Table 1). Mechanistic non-stomatal parameterization remains challenging due to uncertainties in inferred non-stomatal deposition estimates, such as difficulties in separating non-stomatal uptake from soil uptake and in-canopy chemistry (Clifton et al., 2020a). Future evaluation of non-stomatal algorithms requires further measurements such as BVOC emissions and soil moisture (Clifton et al., 2019)."*

***P7 L265:*** *"Simulations using each dry deposition configuration were conducted in the single-site mode in TEMIR for the observational sites listed in Supplementary Table S1."*

***P7 L279:*** *"... For global simulations, the model was run at a spatial resolution of 2°×2.5° driven by MERRA-2 meteorology for each dry deposition configuration. $v_d$ and $G_s$ were summed up by PFT fractions over vegetated land within each grid cell."*

***P9 L323:*** *"... The simulated seasonal average daytime $v_d$ using the different dry deposition schemes..."*

***P9 L336:***

| PFT | Season | Observation | W89 | | | W89FBB | | | W89MED | | | Z03 | | | Z03FBB | | | Z03MED | | |
|---|---|---|---|---|---|---|---|---|---|---|---|---|---|---|---|---|---|---|---|---|
| | | mean±sd | mean±sd | NMBF | NMAEF | mean±sd | NMBF | NMAEF | mean±sd | NMBF | NMAEF | mean±sd | NMBF | NMAEF | mean±sd | NMBF | NMAEF | mean±sd | NMBF | NMAEF |
| DBF | JJA | 0.69±0.10 | 0.90±0.17 | 0.32 | 0.32 | 0.59±0.10 | −0.16 | 0.26 | 0.81±0.24 | 0.18 | 0.41 | 0.55±0.09 | −0.25 | 0.30 | 0.58±0.11 | −0.18 | 0.26 | 0.78±0.25 | 0.14 | 0.41 |
| | MAM | 0.33±0.02 | 0.42±0.13 | 0.27 | 0.43 | 0.28±0.08 | −0.21 | 0.23 | 0.35±0.10 | 0.05 | 0.26 | 0.29±0.08 | −0.13 | 0.27 | 0.31±0.05 | −0.09 | 0.18 | 0.37±0.08 | 0.10 | 0.21 |
| | SON | 0.52±0.20 | 0.49±0.12 | −0.05 | 0.18 | 0.29±0.07 | −0.78 | 0.78 | 0.39±0.13 | −0.34 | 0.34 | 0.41±0.06 | −0.26 | 0.26 | 0.37±0.05 | −0.39 | 0.39 | 0.46±0.11 | −0.11 | 0.13 |
| | DJF | 0.25±0.08 | 0.14±0.05 | −0.86 | 0.97 | 0.14±0.05 | −0.86 | 0.86 | 0.15±0.06 | −0.72 | 0.87 | 0.24±0.04 | −0.04 | 0.21 | 0.26±0.03 | 0.02 | 0.23 | 0.26±0.04 | 0.05 | 0.27 |
| ENF | JJA | 0.58±0.23 | 0.46±0.12 | −0.29 | 0.35 | 0.46±0.11 | −0.30 | 0.42 | 0.47±0.10 | −0.27 | 0.40 | 0.42±0.14 | −0.39 | 0.68 | 0.52±0.14 | −0.14 | 0.44 | 0.53±0.13 | −0.12 | 0.42 |
| | MAM | 0.46±0.15 | 0.35±0.11 | −0.31 | 0.43 | 0.34±0.10 | −0.34 | 0.40 | 0.37±0.12 | −0.24 | 0.37 | 0.42±0.06 | −0.10 | 0.31 | 0.43±0.09 | −0.07 | 0.26 | 0.46±0.11 | −0.01 | 0.26 |
| | SON | 0.47±0.22 | 0.35±0.12 | −0.35 | 0.43 | 0.28±0.07 | −0.64 | 0.68 | 0.26±0.04 | −0.83 | 0.85 | 0.39±0.13 | −0.21 | 0.46 | 0.41±0.15 | −0.13 | 0.37 | 0.40±0.12 | −0.18 | 0.43 |
| | DJF | 0.32±0.21 | 0.17±0.07 | −0.87 | 0.89 | 0.19±0.08 | −0.66 | 0.73 | 0.16±0.06 | −0.98 | 1.01 | 0.30±0.11 | −0.08 | 0.29 | 0.30±0.15 | −0.05 | 0.28 | 0.28±0.12 | −0.14 | 0.36 |
| CRO | / | 0.53±0.16 | 0.50±0.26 | −0.05 | 0.29 | 0.72±0.15 | 0.37 | 0.43 | 0.83±0.13 | 0.54 | 0.54 | 0.54±0.11 | 0.03 | 0.18 | 0.61±0.15 | 0.16 | 0.32 | 0.67±0.14 | 0.27 | 0.32 |
| TRF | / | 0.76±0.48 | 1.11±0.07 | 0.46 | 0.56 | 0.98±0.06 | 0.29 | 0.52 | 1.10±0.10 | 0.44 | 0.53 | 0.47±0.05 | −0.60 | 0.85 | 0.57±0.04 | −0.33 | 0.61 | 0.66±0.07 | −0.14 | 0.48 |
| GRA | JJA | 0.33±0.17 | 0.72±010 | 1.21 | 1.21 | 0.59±0.21 | 0.82 | 0.82 | 0.84±0.28 | 1.56 | 1.56 | 0.50±0.12 | 0.53 | 0.79 | 0.50±0.16 | 0.51 | 0.51 | 0.68±0.21 | 1.08 | 1.08 |
| | MAM | 0.39±0.13 | 0.58±0.13 | 0.48 | 0.48 | 0.43±0.00 | 0.08 | 0.28 | 0.62±0.16 | 0.57 | 0.74 | 0.42±0.11 | 0.06 | 0.48 | 0.46±0.03 | 0.17 | 0.36 | 0.62±0.15 | 0.56 | 0.72 |
| | SON | 0.30±0.06 | 0.59±0.21 | 1.00 | 1.20 | 0.46±0.22 | 0.55 | 0.78 | 0.55±0.26 | 0.88 | 1.03 | 0.42±0.29 | 0.43 | 0.76 | 0.46±0.20 | 0.54 | 0.80 | 0.54±0.22 | 0.82 | 0.95 |
| | DJF | 0.33±0.05 | 0.34±0.26 | 0.02 | 0.68 | 0.24±0.14 | −0.37 | 0.56 | 0.34±0.31 | 0.04 | 0.77 | 0.31±0.15 | −0.08 | 0.46 | 0.35±0.18 | 0.06 | 0.49 | 0.443±0.32 | 0.31 | 0.79 |

***P9 L338:*** *" The dry deposition schemes used in this study fit observed $v_d$ better for deciduous forest and crops. Yet different schemes cannot reproduce daytime $v_d$ well for coniferous forest, grass and rainforest."*

*P9 L350:* *"In previous studies, models mostly underestimated grassland $v_d$ (Hardacre et al., 2015; Pio et al., 2000). Discrepancies between our modeled grassland $v_d$ and previous works mainly arise from the prescribed minimum stomatal resistance ($r_{smin}$) and LAI."*

*P11 L367:* *Figure plotted with larger text size and color blindness friendly palette.*

[Figure]

*P11 L370:* *"... except that for crops where different colors indicate crop types"*

*P12 L382:* *"Non-stomatal $O_3$ deposition includes chemical reactions of $O_3$ with nitric oxide (NO) and biogenic volatile organic compounds (BVOC) from soil emissions (Fares et al., 2012)."*

*P12 L407:* *"Figure 2 shows observed and simulated monthly mean daytime (6:00am~18:00pm) $v_d$ ..."*

*P13 L417:* *"different dry deposition schemes in the following  section"*

*P14 L436:* *"... which is mainly caused by overestimated afternoon  $G_s$."*

**P14 L440:** *"Figure 5 shows the fractions of monthly average daytime stomatal conductance to canopy conductance ($G_c = 1/R_c$), and that higher fractions indicate higher ratios of stomatal deposition to non-stomatal deposition."*

**P16 L461:** *Figure S3 moved to main text as below.*

[Figure]

*Figure 5. Fractions of average monthly daytime stomatal conductance ($G_s$) to canopy conductance ($G_c = 1/R_c$) at the four long-term measurement sites. Black lines indicate fractions calculated with $G_s$ derived using P-M method. Colored solid lines indicate fractions calculated with different dry deposition schemes.*

**P16 L468:** *"The major $O_3$ removal process in the ponderosa pine plantation at the Blodegett Ameriflux site is non-stomatal $O_3$ sink through in-canopy chemical reactions between $O_3$ and BVOC (Fares et al., 2010; Kurpius and Goldstein, 2003)."*

**P17 L511:** *"Z03 considers stomatal blocking that occurs after rain or dew events, and thus simulates lower dry deposition velocities at measurement sites with high precipitation. However, for most observational sites used in this study, precipitation rates are lower than the stomatal blocking threshold throughout the measurement periods, and stomatal blocking contributes little to the differences in simulated $v_d$ across different schemes."*

**P18 L545:** *"...while MED overestimates $G_s$ (NMBF = 0.44), and W89 simulates with NMAEF = 0.41, lower than other schemes (NMAEF > 0.50). Evaluation with long-term measurements in Sect 3.2 finds similar model performance using different stomatal schemes. As shown in Fig. 4a and Fig. 4c, Z03 and FBB simulate comparable diurnal $G_s$ cycles. MED produces higher $G_s$ values than FBB in general."*

**P23 L631:** *"Z03 simulates lower daytime vd than W89 in most regions, except for evergreen needleleaf regions at high latitudes (Fig. 9a), where Z03 simulates higher stomatal deposition than W89 (Fig. 10e). Hence differences in daytime vd for these regions are caused by higher non-stomatal deposition simulated by Z03."*

**P26 L695:** *"… RuBP (ribulose 1,5-bisphosphate)…"*

**P26 L715:** *"… Free-Air CO₂ Enrichment (FACE) experiments…"*

[Figure]

**P27 L733:**
* * *
**List of changes in the Supplementary:**

*"Text S3*

*We use evaporative-resistance form of Penman-Monteith method to keep consistent with SynFlux stomatal conductance. The leaf stomatal conductance is:*

$$g_w^{-1} = \frac{\varepsilon\rho(e_s(T_f)-e)}{pE} - (r_a + r_{b,w}),$$

*where ε is mass ratio between water and dry air, p is air pressure, E is surface moisture flux, $T_f$ is leaf temperature, $e_s(T_f)$ is the saturation vapor pressure at leaf surface. $r_a$ is aerodynamic resistance, $r_{b,w}$ is quasi-laminar layer resistance to water vapor. $T_f$ is estimated as follows:*

$$T_f = T + \frac{H(r_a+r_{b,H})}{c_p\rho},$$

*where T is air temperature, H is sensitive heat, $c_p$ is specific heat of air, ρ is the mass density of air, $r_{b,H}$ is quasi-laminar layer resistance to heat.*

*Stomatal conductance of $O_3$ is calculated with molecular diffusion coefficient ratio 0.6 between $O_3$ and water vapor:*

$$g_s = 0.6g_w \quad "$$

Table S4. List of abbreviations used in this paper with descriptions.

| Symbol | Description |
|---|---|
| $A_n$ | leaf net $CO_2$ assimilation rate |
| BVOC | biogenic volatile organic compounds |
| CLM | Community Land Model |
| CRO | Crop |
| $C_s$ | $CO_2$ concentration at the leaf surface |
| CTMs | chemical transport models |
| DBF | Deciduous Broadleaf Forest |
| $D_i$ | molecular diffusivities for water |
| $DO_3SE$ | The Deposition of $O_3$ for Stomatal Exchange |
| $D_v$ | molecular diffusivities for pollutant gas |
| ENF | Evergreen Needleleaf Forest |
| ESMs | Earth system models |
| FBB | Farquhar-Ball-Berry stomatal scheme |
| $g_0$ | PFT-dependent minimum stomatal conductance |
| $g_{1B}$ | fitted slope parameter for Ball-Berry model |
| $g_{1M}$ | fitted slope parameter for Medlyn model |
| GRA | Grass |
| $G_s$ | Canopy stomatal conductance |

| $h_s$ | leaf surface relative humidity |
|---|---|
| $L$ | Obukhov length |
| LAI | leaf area index |
| $L^{sha}$ | shaded LAI |
| LSMs | land surface models |
| $L^{sun}$ | sunlit LAI |
| MAP | mean annual precipitation |
| MED | Medlyn stomatal scheme |
| MERRA-2 | Modern-Era Respective analysis for Research and Applications version 2 |
| MODIS | Moderate Resolution Imaging Spectroradiometer |
| NMAEF | normalized mean absolute error factor |
| NMBF | normalized mean bias factor |
| NO | nitric oxide |
| $O_3$ | ozone |
| P-M | Penman-Monteith |
| PAR | photosynthetically active radiation |
| PFTs | plant functional types |
| $P_r$ | the Prandtl number for air |
| $R^2$ | $R$-squared value |
| $R_a$ | aerodynamic resistance |
| $R_{ac}$ | in-canopy aerodynamic resistance |
| $R_{adc}$ | lower canopy aerodynamic resistance |
| $R_{ag}$ | ground aerodynamic resistance |
| $R_b$ | quasi-laminar sublayer resistance |
| $r_b$ | leaf boundary resistance |
| $R_c$ | bulk surface resistance |
| $R_c$ | canopy resistance |
| $R_{clx}$ | lower canopy resistance |
| $R_{cut}$ | cuticular resistance |
| $R_{cutd0}$ | reference cuticular resistance for dry condition |
| $R_{cutw0}$ | reference cuticular resistance for wet condition |
| $R_g$ | ground resistance |
| RH | relative humidity |
| $R_s$ | stomatal resistance |
| $r_{smin}$ | minimum stomatal resistance |
| $r_s^{sha}$ | shaded stomatal resistance |

| | |
|---|---|
| $r_s^{sun}$ | sunlit stomatal resistance |
| RuBP | ribulose 1,5-bisphosphate |
| $S_r$ | the Schmidt number |
| SRAD | incoming shortwave solar radiation |
| SW | soil wetness |
| $T$ | surface temperature |
| TEMIR | Terrestrial Ecosystem Model in R |
| TRF | Tropical Rainforest |
| $u_*$ | friction velocity |
| $v_d$ | dry deposition velocity of $O_3$ |
| VPD | vapor pressure deficit |
| W89 | Wesely deposition scheme |
| W89FBB | Wesely deposition scheme replaced with Faquhar-Ball-Berry stomatal scheme |
| W89MED | Wesely deposition scheme replaced with Medlyn stomatal scheme |
| $W_{st}$ | stomatal blocking factor |
| $z$ | reference height |
| $z_0$ | roughness height |
| Z03 | Zhang et al. (2003) deposition scheme |
| Z03FBB | Zhang et al. (2003) deposition scheme replaced with Faquhar-Ball-Berry stomatal scheme |
| Z03MED | Zhang et al. (2003) deposition scheme replaced with Medlyn stomatal scheme |
| $\kappa$ | von Kármán constant |
| $\psi$ | water stress |

**References**

Clifton, O. E., Fiore, A. M., Munger, J. W., & Wehr, R.: Spatiotemporal controls on observed daytime ozone deposition velocity over Northeastern U.S. forests during summer, J. Geophys. Res.-Atmos, 124, 5612– 5628, https://doi.org/10.1029/2018JD029073, 2019.

Liu, X., Tai, A.P.K., Chen, Y. et al. Publisher Correction: Dietary shifts can reduce premature deaths related to particulate matter pollution in China, Nat. Food, https://doi.org/10.1038/s43016-022-00458-2, 2022.

Lombardozzi, D., Levis, S., Bonan, G., Hess, P. G., & Sparks, J. P.: The influence of chronic ozone exposure on global carbon and water cycle, J. Climate, 28(1), 292–305, https://doi.org/10.1175/Jcli-D-14-00223.1, 2015.

Monks, P. S., Archibald, A. T., Colette, A., Cooper, O., Coyle, M., Derwent, R., Fowler, D., Granier, C., Law, K. S., Mills, G. E., Stevenson, D. S., Tarasova, O., Thouret, V., von Schneidemesser, E.,

Sommariva, R., Wild, O., and Williams, M. L.: Tropospheric ozone and its precursors from the urban to the global scale from air quality to short-lived climate forcer, Atmos. Chem. Phys., 15, 8889–8973, https://doi.org/10.5194/acp-15-8889-2015, 2015.

Sadiq, M., Tai, A. P. K., Lombardozzi, D., & Martin, M. V.: Effects of ozone-vegetation coupling on surface ozone air quality via biogeochemical and meteorological feedbacks. Atmos. Chem. Phys., 17(4), 3055-3066. https://doi.org/10.5194/acp-17-3055-2017, 2017.

Sitch, S., Cox, P. M., Collins, W. J., & Huntingford, C.: Indirect radiative forcing of climate change through ozone effects on the land-carbon sink, Nature, 448(7155), 791-U794, http://doi.org/10.1038/nature06059, 2007.

Zhao, Y., Zhang, L., Tai, A. P. K., Chen, Y., and Pan, Y.: Responses of surface ozone air quality to anthropogenic nitrogen deposition in the Northern Hemisphere, Atmos. Chem. Phys., 17, 9781–9796, https://doi.org/10.5194/acp-17-9781-2017, 2017.

Zhou, S. S., Tai, A. P. K., Sun, S. H., Sadiq, M., Heald, C. L., & Geddes, J. A.: Coupling between surface ozone and leaf area index in a chemical transport model: strength of feedback and implications for ozone air quality and vegetation health, Atmos. Chem. Phys., 18(19), 14133-14148, https://doi.org/10.5194/acp-18-14133-2018, 2018.

Zhu, J., Tai, A. P. K., and Hung Lam Yim, S.: Effects of ozone–vegetation interactions on meteorology and air quality in China using a two-way coupled land–atmosphere model, Atmos. Chem. Phys., 22, 765–782, https://doi.org/10.5194/acp-22-765-2022, 2022.